# Role of cell-type specific nucleosome positioning in inducible activation of mammalian promoters

Agata Oruba[1], Simona Saccani[1,2,3 ✉] & Dominic van Essen[1,2,3 ✉]

The organization of nucleosomes across functional genomic elements represents a critical layer of control. Here, we present a strategy for high-resolution nucleosome profiling at selected genomic features, and use this to analyse dynamic nucleosome positioning at inducible and cell-type-specific mammalian promoters. We find that nucleosome patterning at inducible promoters frequently resembles that at active promoters, even before stimulus-driven activation. Accordingly, the nucleosome profile at many inactive inducible promoters is sufficient to predict cell-type-specific responsiveness. Induction of gene expression is generally not associated with major changes to nucleosome patterning, and a subset of inducible promoters can be activated without stable nucleosome depletion from their transcription start sites. These promoters are generally dependent on remodelling enzymes for their inducible activation, and exhibit transient nucleosome depletion only at alleles undergoing transcription initiation. Together, these data reveal how the responsiveness of inducible promoters to activating stimuli is linked to cell-type-specific nucleosome patterning.

[1] Max Planck Institute for Immunobiology & Epigenetics, Stübeweg 51, Freiburg D79108, Germany. [2] Institute for Research on Cancer & Aging, Nice (IRCAN), 28 Avenue Valombrose, Nice 06107, France. [3] These authors contributed equally: Simona Saccani, Dominic van Essen. ✉email: ssaccani@unice.fr; dvanessen@unice.fr

The packaging of eukaryotic genomes into chromatin influences every DNA-templated process. The basic unit of chromatin consists of genomic DNA wrapped around an octamer of histone proteins to form a nucleosome, which alters the conformation and accessibility of the nucleosome-wrapped DNA[1]. At gene promoters, the level of DNA accessibility for binding by transcription factors and by the transcriptional machinery represents a fundamental layer of control[2–4]. Accordingly, promoter nucleosome positioning is known to be highly regulated, and nucleosomes at promoters of actively transcribed genes often occupy precisely defined positions[5–7].

Nucleosome positioning can be assayed by limited enzymatic digestion of chromatin using micrococcal nuclease (MNase) or other enzymes[8–11], which are partially occluded from nucleosome-wrapped DNA and preferentially digest inter-nucleosomal regions[6,12]. A strength of this strategy is that the entire genome is amenable to analysis, and so it can reveal nucleosome positioning at all genomic regions[13–18]. However, this can also represent a drawback, since only a small proportion of analysed DNA corresponds to each genomic position, which limits the coverage and resolution at particular regions of interest. As a consequence, most studies performed in larger genomes have relied on high sequencing depths to provide a snapshot of global nucleosome occupancy, without following changes that occur during stimulation or experimental manipulation.

Studies of nucleosome positioning at promoters in yeast have shown that most actively transcribing genes exhibit a characteristic pattern of nucleosome occupancy, consisting of a region with depleted measured occupancy surrounding the transcription start site (TSS), a consistently positioned nucleosome immediately downstream of the TSS (termed the +1 nucleosome), and regularly spaced—or phased—nucleosomes extending into the transcribed body of the gene[5–7]. A similar behaviour has been found in all metazoan genomes that have been analysed, although in many cases this has been studied as an average of large numbers of genes, which could mask differences betweeen individual promoters[2,19–21].

At stimulus-inducible promoters in mammals, the magnitude of transcription after activation correlates with promoter nucleosome depletion[22], and enzymatic nucleosome remodelling has been shown to play a role in regulating transcription at several well-studied model loci[23–26]. Moreover, it has been shown that many inducible genes that do not require remodelling for their activation are characterised by promoter CpG islands (CGIs), which correlate with higher levels of measured nucleosome depletion[13,15,18,27], suggesting that this may affect their activation requirements[27].

Nevertheless, dynamic changes in nucleosome positioning at mammalian inducible promoters have not been systematically defined at high resolution, and many key questions about its role in gene activation remain unanswered: (i) does the arrangement of nucleosomes at inducible promoters predict their responsiveness, and what changes—if any—occur upon transcriptional activation? (ii) to what extent is promoter nucleosome positioning driven by sequence-encoded features, and, conversely, to what level can it vary according to the cellular context? (iii) does the induction of gene expression establish a stable nucleosomal configuration that is permissive for multiple rounds of transcriptional initiation, or are repeated nucleosome remodelling events required at each cycle of initiation? and (iv) can gene activation occur at all without nucleosome depletion from the transcriptional start site at promoters?

Here, we establish a targeted strategy for mapping nucleosome occupancy at high resolution at defined genomic features, and apply this to investigate its role in gene activation. We find that nucleosome patterning at inducible promoters can define their cell-type-specific responsiveness, and that activation of non-patterned inducible promoters is associated with short-lived nucleosome remodelling events that accompany transcriptional initiation.

## Results

**Development of the ChIP-MNase technique.** To allow analysis of nucleosome positioning at genomic regions-of-interest associated with selected biochemical properties, we developed a protocol that we term 'ChIP-MNase', which couples chromatin immunoprecipitation (ChIP) with on-bead MNase-digestion of the recovered chromatin fragments (summarised in Fig. 1a).

We fixed nucleosomes at their natural positions using formaldehyde cross-linking, and performed ChIP using chromatin fragmented into large 5–10 kb pieces, to allow analysis of kilobase-scale genomic intervals; this also had the beneficial effect of increasing ChIP efficiency, compared to using more conventional sub-kilobase fragments. We then digested immunoprecipitated chromatin using titrated amounts of MNase (Supplementary Fig. 1a, b), and analysed mononucleosomal DNA fragments by high-throughput sequencing.

As a ChIP target to enrich promoters (as well as other gene-regulatory elements), we selected the histone modification H3K4me1. H3K4me1 marks both promoters and enhancers that are active in a particular cell type[28]; however, differently to other H3K4 methylation states, H3K4me1 is also present at many transcriptionally inactive promoters, and at most inducible promoters in the pre-stimulation state (Supplementary Figs. 1c, d and 2a). Fragments mapping to regions surrounding annotated promoters were significantly enriched within the H3K4me1 ChIP-MNase datasets compared to the level corresponding to uniform genomic coverage (or, put differently, the same level of coverage at these regions was attained from fewer sequencing reads, compared to non-targeted MNase digestion). Promoters with enriched coverage included those driving both high- and low-levels of gene expression, as well as promoters of inducible and non-expressed genes, and included a diversity of chromatin and sequence features associated with promoter function (for instance CGIs; Fig. 1b, c, Supplementary Figs. 1c, d and 2a–e). The increased coverage across ChIP-enriched regions compared to other genomic locations, which represents the specificity of the measured signal for alleles bearing the immunoprecipitated mark, was typically around 30-fold. We considered only genomic regions with high sequence coverage for subsequent analyses (Supplementary Fig. 2f, g and see section Methods).

We normalised the ChIP-MNase signal at the kilobase-scale, in order to counteract the uneven recovery of distinct genomic loci by ChIP and remove any contribution of the local profile of the chosen ChIP target across each locus from the MNase digestion profile (Fig. 1d; note that normalisation could also mask any kilobase-scale or promoter-to-promoter variations in overall nucleosome occupancy levels). After normalisation, the ChIP-MNase signal reveals readily discernable protected DNA fragments corresponding to the in vivo positions of nucleosomes (Fig. 1e), which are consistent with profiles obtained by whole-genome MNase digestion using much higher-depth sequencing (confirming that enrichment using H3K4me1 does not lead to aberrant recovery of a non-representative subset of alleles; Supplementary Fig. 3a, b).

The lowered sequencing depth required for ChIP-MNase facilitates inclusion of replicate samples. Previous nucleosome-mapping studies in mammalian genomes have often not included replicates, so nucleosome positions have generally been inferred without any estimate of reproducibility or accuracy[13–15,17,18]. We performed ChIP-MNase using independent replicate samples of two cell-types—3T3 fibroblasts (hereafter fibroblasts) and in

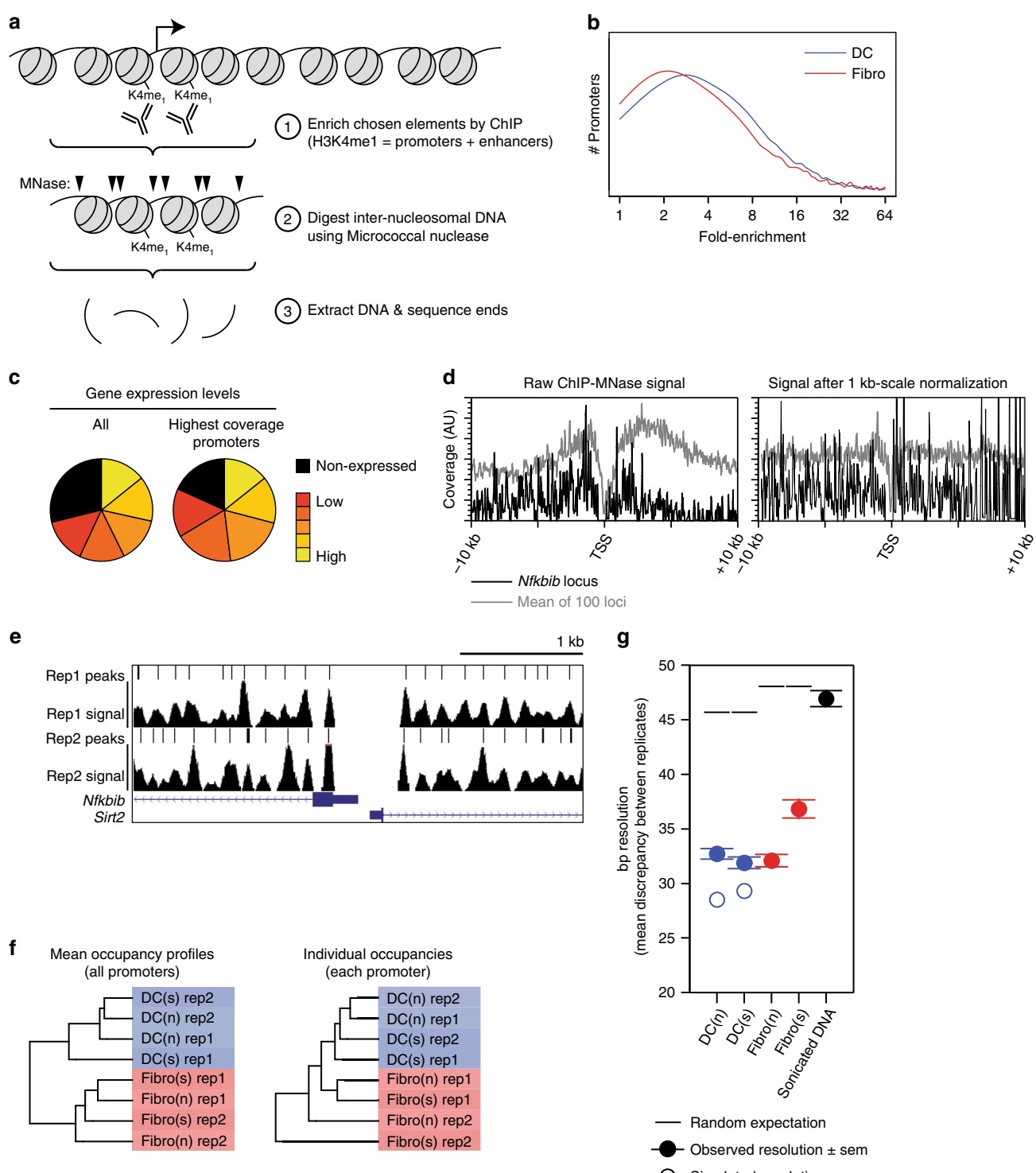

vitro-differentiated dendritic cells (DCs)—under distinct experimental conditions, to measure reproducibility and accuracy. To analyse reproducibility between samples, we used unsupervised hierarchical clustering of the normalised nucleosome occupancy signal surrounding all promoters. Replicate biological samples of each cell-type cluster together (Fig. 1f), demonstrating that any differences in the measured ChIP-MNase profiles arise primarily from biological differences between cell-types, and not from inadvertent differences in sample preparation or treatment. We next estimated preferred nucleosome positions by quantifying occupancy peaks based on their topographic prominence

(see section Methods; Fig. 1e, Supplementary Fig. 3c–e), and determined the accuracy of inferred positions by comparing replicate samples. The mean difference in positions predicted using individual replicate samples is approximately 30 bp (Fig. 1g, Supplementary Fig. 3f, g), confirming the reproducibility of the calculated positions and providing an empirical estimate of the resolution of the technique. Moreover, in silico simulations to eliminate sample-to-sample variation indicate that this figure represents an upper bound arising primarily from analytical limitations, and not from differences between biological samples.

**Fig. 1 High resolution nucleosome mapping at regions-of-interest. a** Outline of ChIP-MNase technique. **b** Distribution of coverage levels attained by ChIP-MNase applied to 10 kb H3K4me1 promoter regions, expressed as the fold-enrichment over the mean level corresponding to homogeneous genomic coverage (equivalent to the fold-reduction in sequencing depth that would be required to attain a fixed level of coverage). **c** Proportion of promoters of non-expressed and expressed genes in fibroblasts among all promoters (left) or among those with the 10% highest levels of coverage after enrichment by H3K4me1 ChIP-MNase. **d** Correction of ChIP-MNase signal for large-scale variation in the magnitude of ChIP recovery. (left) Unprocessed ChIP-MNase signal across an example locus (encompassing the promoter of the *Nfkbib* gene; black), or the mean signal across 100 loci (grey). (right) Processed ChIP-MNase signal after normalisation using a sliding 1 kb window: note that the periodic nucleosomal pattern is retained, while the variation in signal magnitude across and between individual loci is remedied. **e** Example distribution of nucleosome occupancy ('signal') and predicted nucleosome positions ('peaks') across the *Nfkbib* promoter region in replicate ChIP-MNase samples from DCs. **f** Unsupervised hierarchical clustering of nucleosome occupancies from distinct ChIP-MNase samples, applied to mean levels across all promoters (left) or to individual levels at each promoter separately (right). Note that DC and fibroblast samples segregate into distinct clusters (blue and red). **g** Resolution of predicted nucleosome positions, calculated as the mean discrepancy between replicate samples (see for instance panel (**e**)). Solid symbols: observed resolution; open symbols: resolution simulated by random sampling to eliminate biological variation; lines: expected resolution based on random placement with matched mean density. Error bars indicate standard error of the mean (SEM) of $n = 60$ bins; full distributions are shown in Supplementary Fig. 3f. Source data are provided as a source data file.

Together, these results indicate that the ChIP-MNase strategy enables high-resolution mapping of nucleosome occupancies and positions at selected features-of-interest, using substantially lower levels of sequencing than are required by non-targeted approaches.

**ChIP-MNase detects known features of nucleosome positioning.** We first examined general aspects of nucleosome positioning. The genomic distances between mapped locations of nearby MNase-digestion fragments exhibit a periodic pattern, reflecting the typical in vivo spacing of preferred nucleosome positions (Supplementary Fig. 4a–c). The mean internucleosomal distance varies between cell-types: we measured mean spacings of 192 bp in fibroblasts and 183 bp in DCs (Fig. 2a; note that the precision attained by averaging data from millions of mapped fragments greatly surpasses the 30 bp resolution of individual nucleosome positions). This difference in mean nucleosome spacing between cell-types is consistent even when separating loci according to gene expression levels, indicating that it does not arise solely from differences in the proportions of active and inactive loci in each cell type. Previous studies using single samples have reported a similar observation[17,18]. Importantly, however, we find that replicate samples from the same cell-type exhibit highly reproducible spacings, providing confirmation that this represents a true cell-type-specific property, rather than a difference in sample preparation. The mean size of MNase-protected fragments does not differ consistently between cell-types (Supplementary Fig. 4d, e), indicating that the nucleosomal footprint is unchanged, and that differences in nucleosome spacing are caused by different lengths of inter-nucleosomal linker DNA.

We next focused our analysis on gene promoter regions. Mean nucleosome occupancies at active promoters display a highly characteristic pattern (Fig. 2b), with a region of lower mean measured nucleosome occupancy (hereafter NDR for nucleosome depleted [or destabilised] region, see below) immediately upstream of and overlapping the TSS, followed by prominently phased mean occupancy levels downstream of the TSS. The magnitude of the NDR is generally largest at CpG-island (CGI)-containing promoters[13], but it is also apparent at active non-CGI promoters (Fig. 2b). The degree of apparent occpancy at the NDR has been previously shown to depend on the measurement method used, suggesting that it may contain destabilised (or 'fragile') nucleosomes[29–33]), or other particles that confer a reduced footprint of protection from MNase[34–38]. Indeed, using ChIP-MNase, we could readily detect sub-nucleosomal sized (50–100 bp) protected fragments within promoter NDRs (Fig. 2b, Supplementary Fig. 4f). Thus, measured patterning can reflect a combination of nucleosome occupancy levels as well as MNase-accessibility[31,39].

At inactive promoters, which exhibit much-diminished patterning (Fig. 2b), the nucleosomes immediately downstream and upstream from the TSS (the $+1$ and $-1$ nucleosomes) are measurably shifted compared to their average positions at active promoters (Fig. 2c, d), consistent with a mechanistic role in transcription[16,40,41]. Altogether, these observations agree well with many previous data[6,42], and demonstrate that ChIP-MNase is able to detect known features of promoter architecture.

Previous studies of promoter nucleosome positioning, especially in large genomes, have often analysed patterns of averaged occupancy across many promoters, rather than analysing each promoter individually;[15–18] this approach requires lower data coverage and resolution, but it cannot unambiguously reveal aspects of patterning that are affected or obscured by the averaging process. We exploited the high local coverage afforded by ChIP-MNase to define a quantitative measure of patterning at individual promoters, based on their congruence to the mean profile at active promoters (Supplementary Fig. 5a–c), thus enabling statistical comparisons of patterning levels. As expected, this measure confirms that highly patterned promoters are largely absent among non-expressed genes; however, we find that they are still significantly enriched among active genes with low expression levels, as well as among non-CGI promoters (Supplementary Fig. 5d, e).

Mean nucleosome occupancy downstream of the TSS exhibits a gradual loss of the periodic phased profile, which could reflect progressively less-precise ('fuzzy') positioning[6]. However, the same effect can also arise from gradual misalignment of nucleosomes that are nevertheless precisely positioned at each promoter (statistical positioning[43,44]). We used predicted nucleosome positions to realign occupancy profiles at individual promoters, either close to the TSS or at distal locations. We find that the phased profile of mean occupancy is largely undiminished both downstream and upstream of the TSS when nucleosomes are individually aligned (Supplementary Fig. 6a–c), revealing that precisely arrayed nucleosome positioning (or, conversely, their fuzziness) does not appreciably vary at these different promoter positions. However, the phased mean profile remains more evident downstream of the TSS of expressed than of non-expressed genes. We therefore compared the precision of positioning, and the mean and variance of nucleosomal spacing, at active and inactive promoters (Supplementary Fig. 6d–f). Unexpectedly, we found that nucleosome positioning is generally more precise surrounding inactive promoters, with the exception of the nucleosomes immediately adjacent to the TSS. However, the variance of inter-nucleosomal spacing is significantly higher than at active promoters. Thus, the reduced mean phasing at inactive promoters arises largely due to misalignment of unevenly spaced yet precisely positioned nucleosomes. This

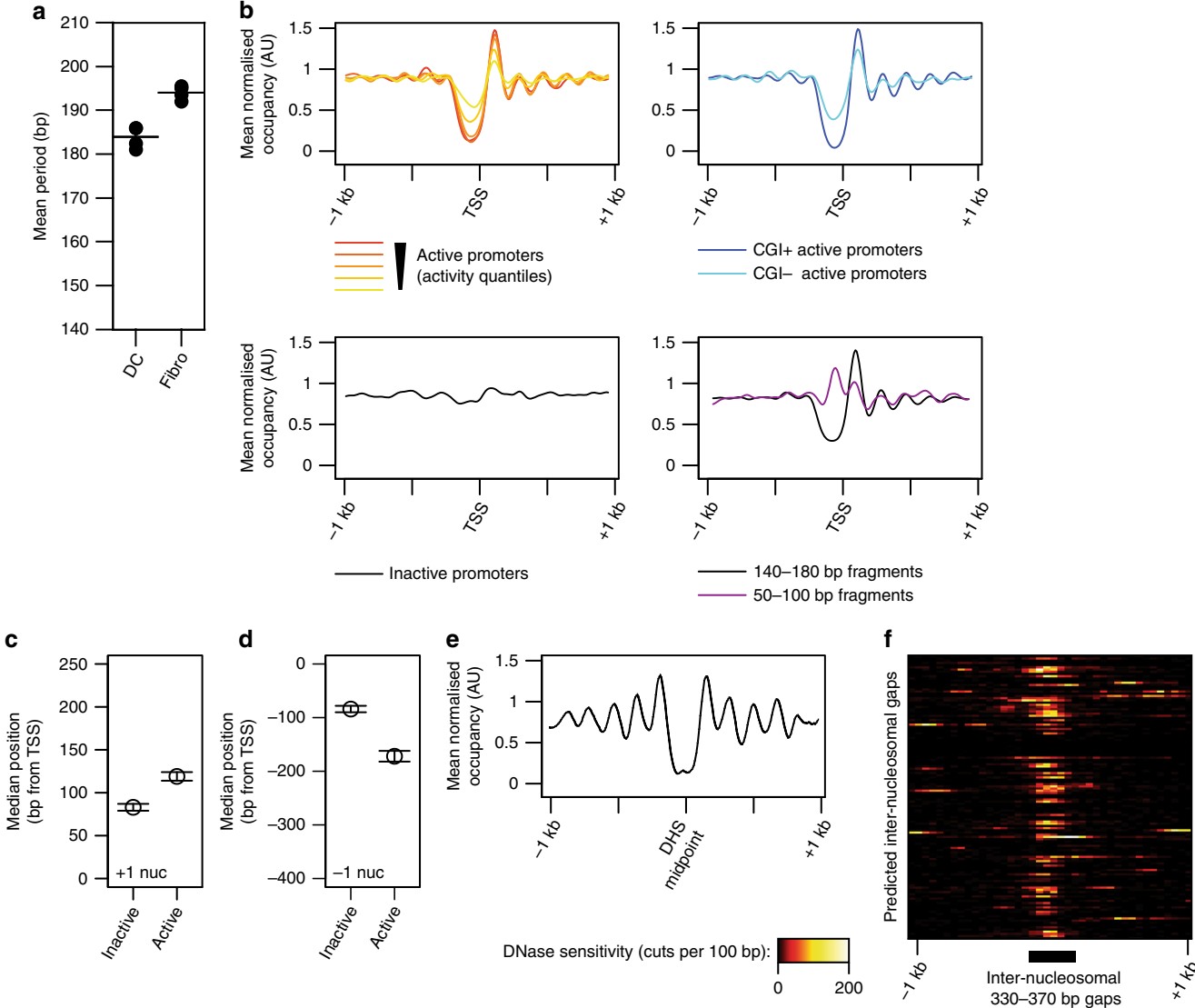

**Fig. 2 Nucleosome positioning at gene-regulatory elements. a** Mean nucleosomal period in replicate ChIP-MNase samples from DCs and fibroblasts. Dots indicate replicate samples; lines denote mean values. **b** Profiles of mean nucleosomal occupancy surrounding annotated gene promoters grouped by quantiles of transcript levels of expressed genes ('active promoters'; top left), by presence/absence of annotated CpG islands at promoters of expressed genes (top right), at promoters of non-expressed genes ('inactive promoters', bottom left), or comparing the coverage by sub-nucleosomal (50–100 bp) and mononucleosomal (140–180b) DNA fragments (bottom right). **c**, **d** Median predicted positions of downstream ('+1 nuc'; (**c**)) or upstream ('−1 nuc'; (**d**)) nucleosome midpoints at inactive and active gene promoters. Positive distances indicate positions downstream of the TSS. Error bars indicate 99% confidence intervals ($n = 3209$ inactive; 1987 active). Source data are provided as a source data file. **e** Profiles of mean nucleosomal occupancy in fibroblasts surrounding midpoints of encode-annotated DHS regions. **f** DNase sensitivity surrounding individual predicted inter-nucleosomal gaps of 330–370 bp that are consistent between 2 replicate ChIP-MNase samples in fibroblasts: 39.5% of all predicted gaps are associated with mean DNase sensitivities of ≥10 cuts per 100 bp.

finding ties in with recent data showing that periodically spaced nucleosome arrays in Drosophila are selectively enriched at active promoters[45].

We also analysed nucleosome positioning at non-promoter regulatory elements identified by DNase hypersensitivity. Mean nucleosome occupancy is depleted at DNase-hypersensitive sites (DHS[42,46]; Fig. 2e). Notably, although a diverse variety of processes can contribute to DHS formation, most sites are characterised by a uniformly sized gap consistent with depletion of a single nucleosome. Accordingly, by computationally searching for single-nucleosome-sized gaps outside promoter regions, we find that up to 40% of these are marked by elevated DNase sensitivity (Fig. 2f). This suggests that despite the biochemical heterogeneity of proteins that associate

with DHSs, these sites are characterised by a strikingly consistent nucleosomal arrangement.

Collectively, these global analyses confirm that ChIP-MNase is readily able to recapitulate known aspects of nucleosome positioning at individual genomic sites, and that the high resolution afforded at specific targeted regions can reveal additional insights.

**Nucleosome patterning defines two inducible promoter classes.** We next analysed nucleosome positioning during activation of stimulus-inducible promoters. The high resolution of ChIP-MNase is instrumental when analysing minor subsets of promoters, for which large numbers of instances are not available to generate averaged profiles from lower-coverage data. We treated

DCs with bacterial lipopolysaccharide (LPS), a potent pro-inflammatory stimulus that activates rapid expression of hundreds of target genes, and identified LPS-inducible genes by transcriptomic analysis of mRNA levels (note that genes assigned as non-inducible in this setup may nevertheless include some that might be induced by other stimuli). We then analysed nucleosome positioning in ChIP-MNase datasets from replicate samples before and after cellular stimulation.

As described above, the mean nucleosomal occupancy across active promoters follows a characteristic pattern. Quantitative analysis of individual promoters indicates that this patterning is shared by the majority (>80%) of active promoters, and that it is absent from almost all inactive promoters (Fig. 3a–c). Strikingly, we observed that nucleosome patterning at many—but not all—inducible promoters closely resembles that at active promoters, even in non-stimulated cells before induction of gene expression. This includes the presence of a clear NDR upstream of the TSS, as well as precisely positioned TSS-adjacent nucleosomes and less-precise but consistently spaced downstream nucleosomes (Fig. 3a, b, Supplementary Fig. 7a–f). Patterning of inducible promoters is not restricted to a subpopulation of promoters that are transcriptionally active in the non-stimulated state (Supplementary Fig. 7e, f), nor does it arise through preferential recovery of a minority of patterned alleles by the H3K4me1 ChIP-MNase technique (Supplementary Fig. 8a–c). Unsupervised hierarchical clustering of both mean and individual promoter nucleosome occupancy levels confirmed that overall nucleosome positioning at inducible promoters more-closely matches that at active than at inactive promoters (Fig. 3c, Supplementary Fig. 7g).

However, quantification of nucleosome patterning at individual promoters revealed that despite their bulk similarity to active promoters, inducible promoters are heterogeneous and can be divided into two groups: 'patterned' inducible promoters display features of active promoters even before stimulus-driven activation, including a pre-formed NDR at the future TSS; in contrast, 'non-patterned' inducible promoters lack a clearly measurable NDR and conform better to the arrangement of steady-state inactive pomoters (Fig. 3c, e, f). At some inducible promoters, low basal levels of pre-stimulation transcription may contribute to patterning; however, similar levels can also be detected at many non-patterned inducible promoters, and patterning and pre-stimulation transciption levels are not strongly correlated (Fig. 3f, Supplementary Fig. 7e, f). Likewise, although the fraction of CGI promoters is higher among those of inducible than of non-expressed genes, many patterned inducible promoters lack a CGI, indicating that—similarly to active promoters—patterning is not invariably linked to the presence of CGIs (Supplementary Fig. 8d, e).

Unexpectedly, nucleosome patterning at both patterned and non-patterned inducible promoters was largely unchanged upon stimulus-driven activation. Patterned inducible promoters, which already resemble active promoters in the unstimulated state, mostly retained this nucleosomal patterning after stimulation, and the subset of inducible promoters that did not display this patterning before stimulation also generally did not acquire it afterwards (Fig. 3e, f). Thus, stimulus-induced activation of transcription can occur without requiring, or driving, major changes to promoter nucleosome patterning.

**Nucleosome patterning can predict promoter responsiveness**. The presence of an NDR and pre-patterned nucleosomes at some stimulus-inducible promoters could arise from inherent, constitutive positioning signals encoded in their DNA sequence[7,13,18,47–53]; however, an alternative possibility is that nucleosomal patterning is independently established in each

cell type, and could thus identify the specific set of promoters that are stimulus responsive. To address this, we investigated the behaviour of promoters that are LPS-inducible in DCs, when they are instead exposed to a different stimulus in a different cell type. We stimulated fibroblasts with tumour necrosis factor alpha (TNF-α), an endogenous inflammatory cytokine that—like LPS—activates the NF kappa B pathway, and that induces an overlapping but distinct set of target genes.

We first classified DC-inducible promoters into sets according to whether they were active, inducible or inactive in fibroblasts, and examined nucleosome patterning in each set separately (Fig. 4a). In DCs, all three sets were heterogeneous, and each contained a similar mixture of patterned and non-patterned promoters (Figs. 4a and S9a). However, when we examined nucleosome occupancies at the same promoters in fibroblasts, we found marked differences between distinct sets of promoters. DC-inducible promoters that are active or inducible in fibroblasts exhibited a similar heterogeneity of nucleosome patterning levels. Thus, as in DCs, the nucleosome patterning of many fibroblast-inducible promoters resembles that of active promoters. In contrast, DC-inducible promoters that are inactive (and non-inducible) in fibroblasts uniformly lack any patterning, and do not exhibit any measurable NDR, despite high ChIP-MNase coverage and resolution at these promoters (Fig. 4a, Supplementary Fig. 9a, d). Quantitation of patterning levels confirmed that this difference is highly significant ($p \leq 3.7 \times 10^{-4}$). Thus, nucleosome patterning at inducible promoters can be cell-type-specific, and can reflect the stimulus-responsiveness of each promoter in a given cell type. This also indicates that the pattern of nucleosome positions at many inducible promoters—including the pre-stimulation presence of an NDR—is not universally specified by their DNA sequence alone (although this may nevertheless be true for some).

The resolution and reproducibility afforded by ChIP-MNase prompted us to test whether the measured nucleosome occupancies at each individual promoter could be sufficient to prospectively sort them and predict aspects of their behaviour (rather than retrospectively analysing nucleosome patterning after sorting based on their observed behaviour). We began by using hierarchical clustering of individual DC-inducible promoters, based on their patterns of nucleosome occupancies in DCs and fibroblasts: this revealed three principal clusters, which were distinguishable by the presence or absence of an NDR in each cell type (Fig. 4b, Supplementary Fig. 9b). Notably, and in line with our findings above, promoters that are inactive and non-inducible in fibroblasts are almost completely excluded from the cluster of NDR-containing promoters in fibroblasts, and are highly enriched within the cluster of promoters that contain an NDR only in DCs ($p \leq 1.6 \times 10^{-3}$). Thus, nucleosome patterning is sufficient to predict the cell-type specific behaviour of some promoters. Conversely, promoters that are inducible in fibroblasts were not distinguishable from those that are active by this clustering, in agreement with the similarity of their patterning when they are analysed as separate groups (Figs. 3a and 4a).

The unsupervised nature of hierarchical clustering provides an unbiased indication of which promoters behave similarly, but it does not readily indicate the particular aspect(s) of the data that most-strongly drive the clustering. We therefore tested whether one could recapitulate the same level of prediction by simple grouping based on the quantified level of nucleosome patterning at each individual promoter (as Fig. 3e). Indeed, this approach was able to separate active/inducible from inactive promoters in fibroblasts with a similar level of discrimination to that of unsupervised clustering ($p \leq 2.1 \times 10^{-5}$; Fig. 4c, Supplementary Fig. 9c), confirming that the level of nucleosome patterning defined here is a major predictive determinant. Moreover, when

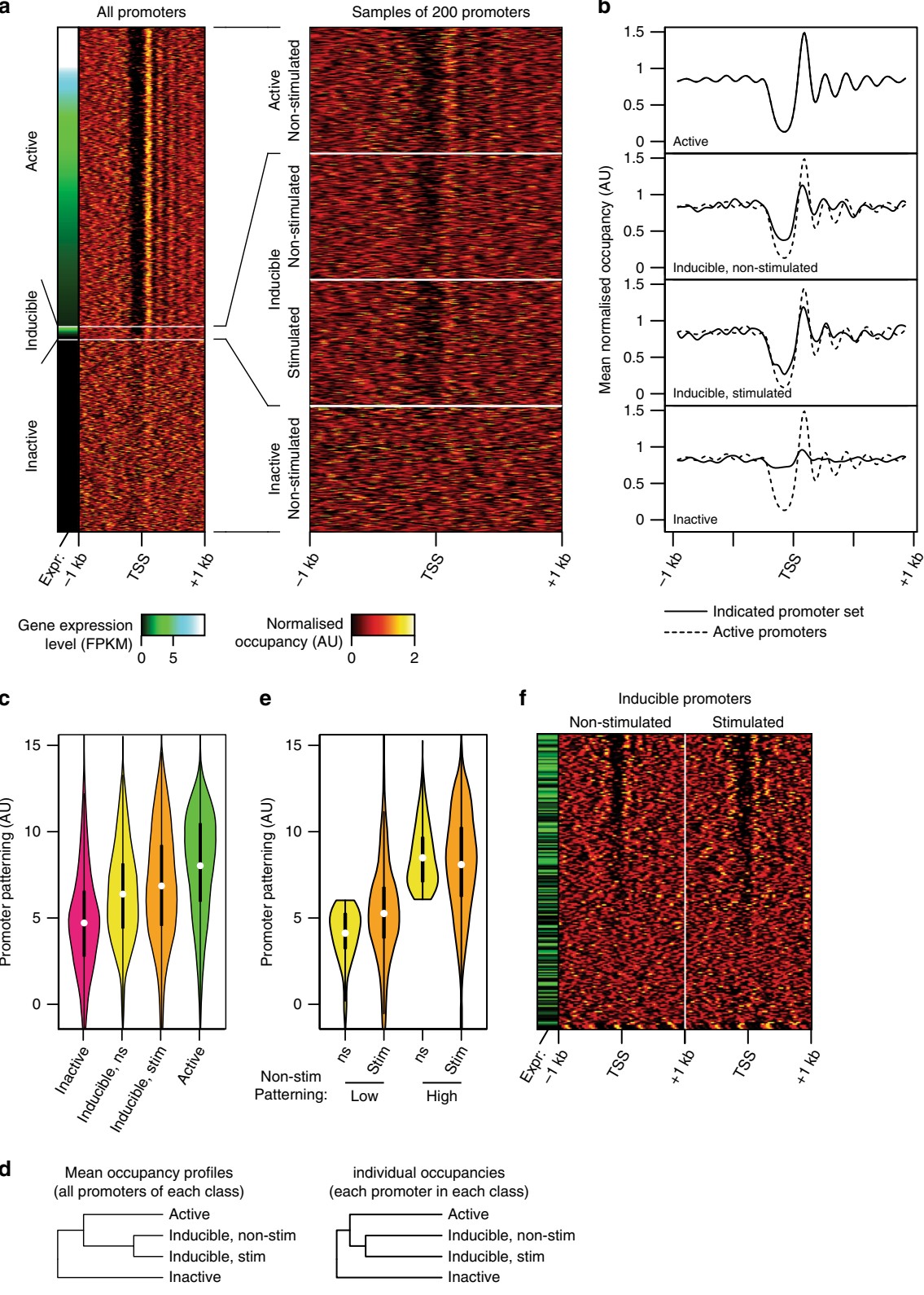

considering those promoters that exhibit high levels of nucleosome patterning in DCs, the level of patterning in fibroblasts before stimulation was strongly indicative of their subsequent responsiveness, and was alone sufficient to predict the future behaviour of more than 80% of fibroblast-inducible promoters (Supplementary Fig. 10). Inducible promoters were again indistinguishable from active promoters in these analyses,

although we cannot exclude that other, more-subtle aspects of nucleosome positioning or accessibility might be able to differentiate them. In each of these analyses, we observed that patterned, NDR-containing DC-inducible promoters consistenty retain their patterning in fibroblasts whenever they are active or inducible in fibroblasts, but that their patterning is lost when they are inactive. On the other hand, non-patterned DC-inducible

**Fig. 3 Pre-stimulation nucleosome patterning at inducible promoters. a** Left: heatmap of nucleosome occupancy levels surrounding active (top; $n =$ 6695), inducible (middle; $n = 214$) and inactive (bottom, $n = 3225$) promoters in DCs; sidebar indicates corresponding gene expression levels (in non-stimulated cells). Right: expanded heatmaps of representative samples of 200 promoters. **b** Profiles of mean occupancy levels in DCs, for active promoters (top), inducible promoters in non-stimulated (upper middle) and LPS-stimulated (lower middle) cells, and at inactive promoters (bottom). Dotted lines indicate the mean level at all active promoters for comparison. **c** Violin plots of quantified promoter patterning levels at inactive, inducible and active promoters in DCs. Totally, 84% of active promoters exhibit patterning levels greater than the median level of inactive promoters, while 86% of inactive promoters have levels less than the median of active promoters. Significance of difference in patterning to inactive promoters: inducible, non-stimulated (ns) $p = 1.1 \times 10^{-25}$; inducible, stimulated $p = 2.8 \times 10^{-34}$; active $p = 1.7 \times 10^{-218}$; difference to active promoters: inducible, ns $p = 2.2 \times 10^{-26}$; inducible, stim $p = 4.3 \times 10^{-12}$ (two-tailed Mann–Whitney $U$ test). Thick bars indicate limits of quartiles; dots indicate means. **d** Hierarchical clustering of nucleosome occupances from distinct sets of promoters, applied to mean levels across all promoters (left) or to individual levels at each promoter separately (right). **e** Violin plots of quantified promoter patterning levels at inducible promoters in DCs, grouped by patterning levels in non-stimulated cells. Totally, 86% of promoters that are unpatterned in non-stimulated DCs retain patterning levels upon stimulation less than the median level of those that are patterned in non-stimulated DCs, while 89% of patterned promoters in non-stimulated DCs retain patterning levels upon stimulation greater than the median of those unpatterned in non-stimulated DCs. Significance of difference in patterning between groups after stimulation (orange violins): $p = 2.2 \times 10^{-30}$ (two-tailed Mann–Whitney $U$ test). Thick bars indicate limits of quartiles; dots indicate means. **f** Heatmap of nucleosome occupancy levels surrounding inducible promoters in non-stimulated (left) and LPS-stimulated (right) DCs (same data as panel (**a**), here sorted by quantified patterning levels, to reveal the heterogeneous patterning of inducible promoters). The sidebar indicates gene expression levels in non-stimulated cells, illustrating that low-level transcription can be detected at promoters with both high and low levels of patterning.

promoters generally remain non-patterned in fibroblasts, irrespectively of their activity in this cell type (Fig. 4c, e, Supplementary Fig. 9e, f). Thus, the cell-type specific inducibility of some promoters is invariably linked to their nucleosomal patterning, whereas other inducible promoters are never stably associated with the patterned configuration. This suggests that these two classes of promoters may be functionally distinct, or controlled by different processes and/or factors (see later).

These analyses also revealed that promoters that are consistently patterned in both cell-types tested, which correspond to those that are always active or inducible, are strongly enriched for CGI-containing promoters (Fig. 4d). Thus, the presence of a promtoter CGI is associated with broadly active or broadly inducible genes, and of non-cell-type-specific promoter patterning; it may thus represent a determinant of active patterning and NDR formation at this subset of promoters.

**Patterned inducible promoters are poised for activation.** The patterning observed at some inducible promoters before stimulation implies that their responsiveness may already be pre-configured into their nucleosomal arrangement. In particular, the presence of a pre-stimulation NDR could facilitate stimulus-inducible activity in a particular cell-type by rendering the TSS accessible for binding by the transcriptional machinery (such as the pre-initiation complex [PIC] and RNA pol-II[12]). This was proposed as a likely prerequisite for inducible activation of CGI promoters[27]—although until now it has not been assessed at the level of in vivo nucleosome positioning—and our data suggest that this may also be a requirement for patterned promoters that lack a CGI. To explore this further, we directly analysed RNA pol-II levels at inducible genes in non-stimulated and stimulated fibroblasts.

Inducible genes with patterned promoter nucleosomes are strongly enriched for the presence of promoter-bound, paused RNA pol-II in non-stimulated cells (Supplementary Figs. 11 and 12a). Paused RNA pol-II could also be detected at a smaller fraction of non-patterned inducible promoters, indicating that although its binding may contribute to the promoter nucleosomal configuration[54], it is not alone sufficient to drive the observed division of patterned and non-patterned promoters. We also observed that patterned inducible promoters are enriched in non-stimulated cells for the presence of histone modifications associated with gene activation, including H3K4me2 and H3K4me3 (Supplementary Fig. 12b). Thus, nucleosome patterning at inducible promoters is generally coupled with pre-loading

of RNA pol-II before stimulation and histone marks that together imply that they are poised for activation[22,27,54–58].

The presence of paused RNA pol-II at patterned inducible promoters might enable rapid activation without requiring stimulus-driven polymerase recruitment[55]. To investigate this, we examined RNA pol-II at promoters and in the transcribed regions of inducible genes, before and after stimulus-induced activation. Inducible genes with both patterned and non-patterned promoters exhibited strong increases in RNA pol-II levels upon stimulation, not only along their gene bodies but also at promoter regions. Moreover, by evaluating changes at individual promoters separately, we observed that even promoters with high levels of pre-stimulation polymerase underwent further increases after stimulation (Supplementary Fig. 12c, d). Hence, inducible gene expression from patterned promoters can reflect both the activation of pre-bound, paused RNA pol-II, as well as stimulus-induced transcriptional initiation.

**BAF-driven activation of non-patterned inducible promoters.** Our finding that most non-patterned promoters are inducibly activated without any measurable NDR was unexpected, and raises the question of how RNA pol-II or other components of the transcriptional machinery can gain access to the TSS at these promoters. Previous studies have shown that the BAF nucleosome-remodelling complex (also known as SWI/SNF) is required for activation of many stimulus-inducible genes[24,27], and it has been proposed that this complex is involved in clearing nucleosomes from promoters[59,60]. Moreover, it has been shown that CGI-containing inducible promoters, which are often strongly patterned with robust NDRs (Fig. 4d[13]), are generally not BAF dependent[27]. Together, these data have led to a well-accepted model whereby the BAF complex is required to establish accessibility at many inducible promoters upon activation[27].

A simple expectation of this model would be that the BAF complex may remodel nucleosomes at inducible promoters from a non-permissive, inactive arrangement to a permissive, NDR-containing arrangement resembling the patterning of other active promoters. Thus, our finding that activation of non-patterned inducible promoters is not associated with establishment of a stable, patterned configuration was unanticipated (Figs. 3e, f and 4). We therefore directly investigated whether the nucleosome remodelling activity of the BAF complex is required to initiate transcription from non-patterned inducible promoters.

We used shRNA to stably knock-down the mRNAs encoding the two catalytic remodelling subunits of the BAF complex, Brm

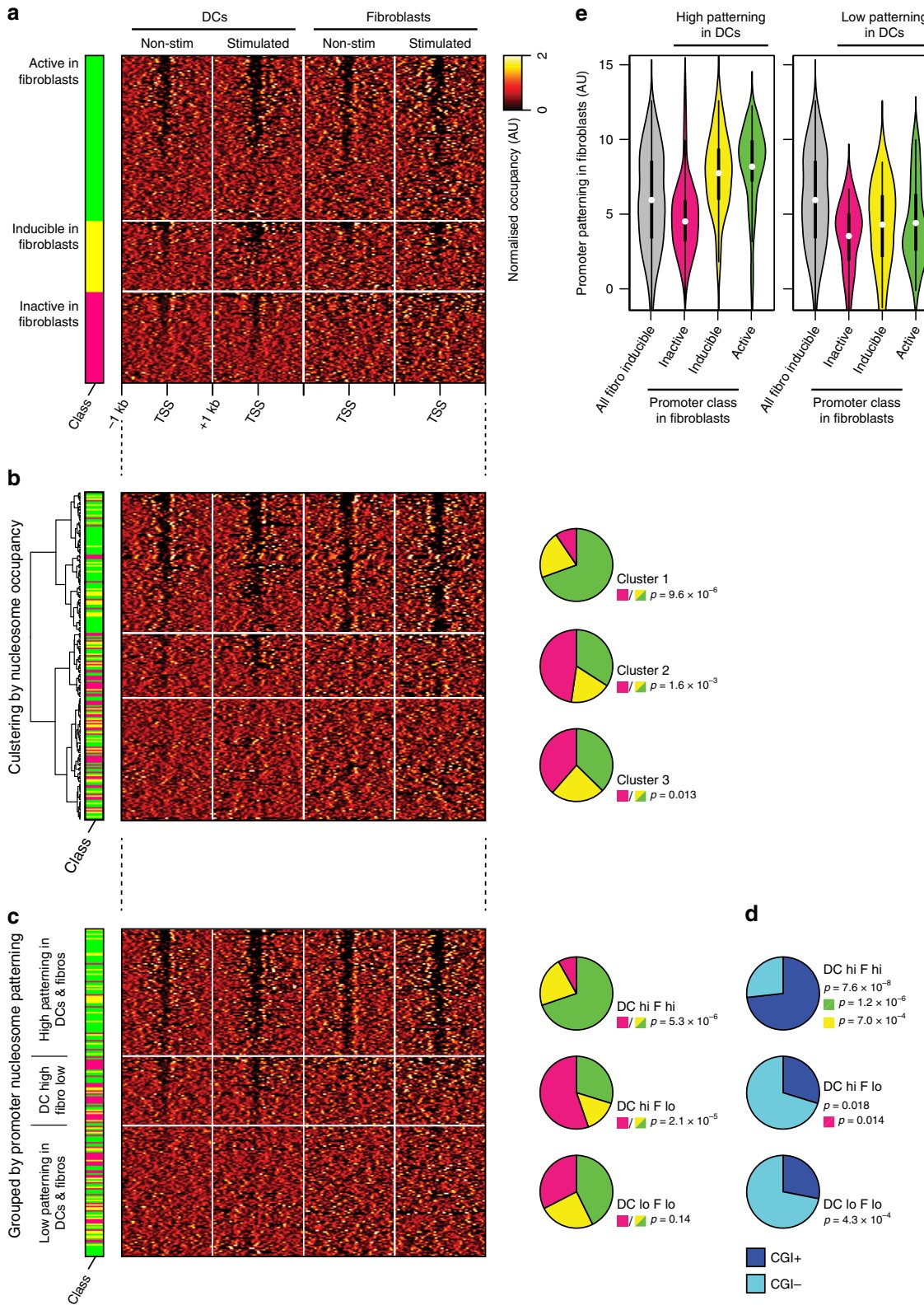

and Brg1, in fibroblasts. A single shRNA targeting a shared sequence was able to reduce levels of both transcripts by around fivefold (Fig. 5a[24]). We then measured gene expression levels and promoter nucleosome occupancies in Brm/Brg1-double-knock-down fibroblasts (hereafter BAF knockdown).

Expression levels of most genes were undisturbed in BAF-knockdown cells, implying that the activity of the BAF complex is non-essential for the majority of steady-state transcription (or that the low residual levels in knockdown cells are sufficient; Fig. 5b). Similarly, nucleosome occupancies and patterning at promoters of genes that were both expressed and non-expressed under steady-state conditions were largely unchanged in BAF-knockdown cells and the presence or absence of an NDR was generally unaffected by BAF knockdown (Fig. 5c, Supplementary

**Fig. 4 Cell-type-specific promoter nucleosome patterning. a** Heatmap of nucleosome occupancy levels surrounding promoters that are inducible in DCs, in non-stimulated and LPS-stimulated DCs (left panels) and in non-stimulated and TNF-α-stimulated fibroblasts (right panels). Promoters are grouped by their acvity class in fibroblasts, indicated by sidebar: active (top, green), inducible (middle, yellow) or inactive (bottom, magenta), and sorted by their level of patterning. **b** Left: heatmap of nucleosome occupancy levels as panel a, sorted by hierarchical clustering of weighted nucleosome profiles. The three highest-level clusters are indicated. Right: pie charts summarising the proportion of each promoter class in each cluster, coloured as sidebar in panel (**a**). p Values indicate the significance of enrichment or depletion of promoters that are inactive in fibroblasts compared to those that are active or inducible (two-tailed binomial test). Active and inducible promoters were not statistically distinguishable (p > 0.05 for all clusters). **c** Left: heatmap of nucleosome occupancy levels as panel (**a**), grouped by promoter nucleosome patterning levels. Right: pie charts summarising the proportion of each promoter class in each group, as panel (**b**). **d** Pie charts summarising the proportion of CGI+ and CGI− promoters in each group from panel (**c**). p Values indicate the significance of enrichment of CGI+ promoters in each group among all promoter classes, or calculated separately only among the subsets that are active (green; first group), inducible (yellow; first group) or inactive (red; second group) in fibroblasts (two-tailed binomial test). **e** Violin plots of quantified promoter patterning levels in fibroblasts, at promoters that are inducible and that exhibit high (left panel, $n = 134$) or low (right panel, $n = 90$) nucleosome patterning levels in DCs, compared all promoters that are inducible in fibroblasts (grey violins). At promoters that are inducible and patterned in DCs, high patterning in fibroblasts predicts active or inducible promoters (left). At promoters that are inducible but non-patterned in DCs, fibroblast patterning is always low and does not predict activity (right). Significances of differences to inactive promoters: high DC patterning, fibroblast-inducible: $p = 7.0 \times 10^{-5}$; fibroblast-active: $p = 1.2 \times 10^{-7}$; low DC patterning, fibroblast-inducible: $p = 0.30$; fibroblast-active: $p = 0.99$ (two-tailed Mann–Whitney $U$ test). Thick bars indicate limits of quartiles; dots indicate means.

Fig. 13a, b). Therefore, establishment and maintenance of a stable promoter NDR at most active genes does not require normal levels of remodelling activity by the BAF complex. In contrast, stimulus-inducible genes, and genes with cell-type dependent expression, are significantly enriched for BAF-dependent activation, with up to one-third of inducible genes exhibiting disrupted expression in BAF-knockdown fibroblasts (Fig. 5b). This confirms that the level of knock-down is sufficient to reveal BAF-dependent transcription, and further indicates that BAF-driven nucleosome remodelling is particularly required for regulated—rather than ubiquitous—gene activation. However, not all inducible genes are associated with BAF-dependent expression: in particular, activation of inducible genes bearing patterned promoters was mostly unaffected by BAF-knockdown (Fig. 5d, Supplementary Fig. 13c–e). These promoters generally retained their patterning in BAF knockdown cells, including the presence of an NDR. Thus, promoters with a measurably nucleosome-depleted TSS before stimulation are generally amenable to full transcriptional activation without BAF-driven remodelling.

These findings are consistent with the notion that a major role of the BAF remodelling complex is to enable or facilitate inducible transcription from promoters at which the TSS is normally occluded by nucleosomes. In agreement with this hypothesis, we found that inducible genes with non-patterned promoters, which lack a stable pre-stimulation NDR, are highly enriched for BAF-dependent activation (Fig. 5d, Supplementary Fig. 13c–e). As previously reported[27], these include many non-CGI promoters. However, BAF-dependent activation of non-patterned inducible promoters is not accompanied by establishment of a stable NDR or conversion to a patterned nucleosome arrangement, and the absence of a detectable NDR is unchanged in BAF-knockdown cells (within the time-frame analysed; Fig. 5d, e, Supplementary Fig. 13d). BAF knock-down also did not impair inducible recruitment of the transcriptional activator NF kappa B p65 to target sites at TNF-α-responsive promoters (Supplementary Fig. 14).

Using publicly available datasets of genome-wide binding by the BAF complex in fibroblasts[61,62], we found that in unstimulated cells, the BAF complex binds to promoter regions of the majority of both expressed and inducible genes (Supplementary Fig. 15). Furthermore, at inducible genes, BAF complex binding is detectable before stimulation at both patterned and non-patterned promoters, despite their different remodelling requirements for activation. It is not known how the BAF complex is generally recruited to its genomic target sites, but these data imply that it broadly binds to active and stimulus-inducible promoters independently of their remodelling requirements.

**Transient remodelling at initiating non-patterned promoters.** The finding that the BAF nucleosome remodelling complex is specifically required for activation of non-patterned inducible promoters provides direct in vivo support for a long-standing model for inducible gene activation[27]. In particular, it corroborates the premise that BAF-independent activation of CGI promoters can be explained by the presence of a pre-formed NDR. However, our finding that BAF-dependent activation of unpatterned promoters is not accompanied by establishment of a stable NDR raises the question of whether nucleosome remodelling may be restricted to a subset of alleles or specific step(s) of transcriptional activation.

The measured nucleosome occupancy at each promoter represents the mean of both alleles, in samples prepared from millions of cells. Thus, we first considered whether the lack of a detectable NDR could be explained if only a minor fraction of promoters are activated under our stimulation conditions, while the majority of alleles remain inactive.

In addition to enriching for specific genomic regions, ChIP-MNase can selectively enrich sub-populations of alleles if they are marked by a biochemical property that can be chosen as a target for ChIP. Therefore, to specifically analyse only transcriptionally active alleles, we performed ChIP-MNase using antibodies directed against the phosphorylated form of serine 7 in the C-terminal domain repeats of RNA pol-II (pol-II S7P). This modification is associated with the elongating form of RNA pol-II[63] and thus selectively marks the alleles of genes that are being actively transcribed (Supplementary Fig. 16a–c). Nucleosome profiles in TNF-α-stimulated fibroblasts, revealed by pol-II S7P ChIP-MNase, clearly indicate that the actively-transcribing alleles of inducible genes nonetheless include unpatterned promoters lacking a measurable NDR (Fig. 6b, d, Supplementary Fig. 16d), and that these account for the majority of inducible promoters that were categorised as unpatterned based on analysis of all H3K4me1-marked alleles. Thus, BAF-dependent activation of unpatterned inducible promoters does not entail formation of a stable NDR at the TSS of transcribing alleles.

The set of actively transcribing alleles of a particular gene within a population can contain instances at which transcriptional initiation is ongoing, but also other instances at which elongating RNA pol-II has already cleared the promoter region. Hence, although the absence of a measurable NDR at transcribing alleles could be consistent with transcriptional initiation from a

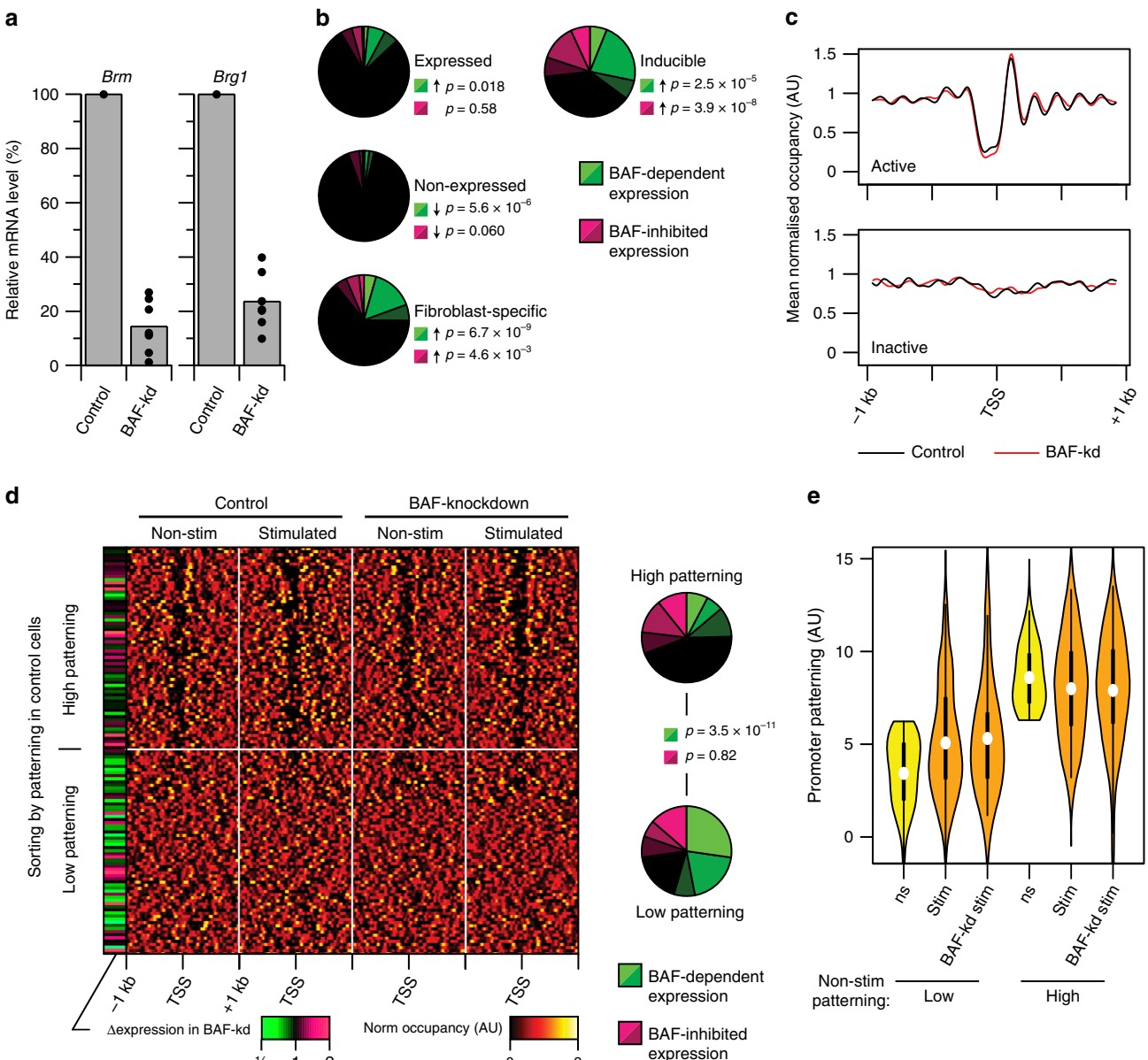

**Fig. 5 BAF-driven activation of non-patterned inducible promoters. a** Levels of *Brm* (left) and *Brg1* (right) mRNA levels in control and BAF-knockdown fibroblasts. Dots indicate measurements from replicate cultures. Source data are provided as a source data file. **b** Pie charts summarising the proportion of genes with BAF-dependent (green; mRNA levels decreased in BAF-kd cells) or BAF-inhibited (magenta; mRNA levels increased in BAF-kd cells) expression, among those that are expressed (top left), non-expressed (middle left) or inducible (top right) in fibroblasts, or that are expressed in fibroblasts and non-expressed in DCs ('fibroblast-specific'; bottom left). *p* Values indicate the significance of enrichment (upward arrow) or depletion (downward arrow) within each group (binomial test). **c** Profiles of mean nucleosomal occupancy surrounding active (top) or inactive (bottom) promoters, in control (black) or BAF-knockdown (red) fibroblasts. **d** Left: heatmap of nucleosome occupancy levels surrounding promoters that are inducible in fibroblasts, in non-stimulated and TNF-α-stimulated conrol fibroblasts (left panels) and in BAF-knockdown fibroblasts (right panels). Promoters are grouped by nucleosome patterning levels in non-stimulated control cells. The sidebar indicates the level that inducible gene expression from each promoter is BAF dependent or inhibited. Right: pie charts summarising the proportion of genes with BAF-dependent (green) or BAF-inhibited (magenta) expression levels in each group. *p* Values indicate the significance of enrichment of BAF-dependent promoters among those with low nucleosomal patterning levels (binomial test). **e** Violin plots of quantified promoter patterning levels at inducible promoters in fibroblasts, grouped by patterning levels in non-stimulated cells. Among promoters that are non-patterned in non-stimulated fibroblasts, 83% (in control cells) and 89% (in BAF-knockdown cells) retain patterning levels upon stimulation less than the median level of those that are patterned in non-stimulated cells, while among promoters that are patterned in stimulated fibroblasts, 97% (control) and 94% (BAF-kd) retain patterning levels upon stimulation greater than the median of those unpatterned in non-stimulated DCs. Significance of differences between low patterning and high patterning promoters after stimulation: control: $p = 1.3 \times 10^{-8}$; BAF-kd: $p = 1.5 \times 10^{-8}$; levels of control and BAF-kd not significantly different from each other in either case: low patterning promoters: $p = 0.79$; high patterning promoters $p = 0.69$ (two-tailed Mann–Whitney *U* test). Thick bars indicate limits of quartiles; dots indicate means.

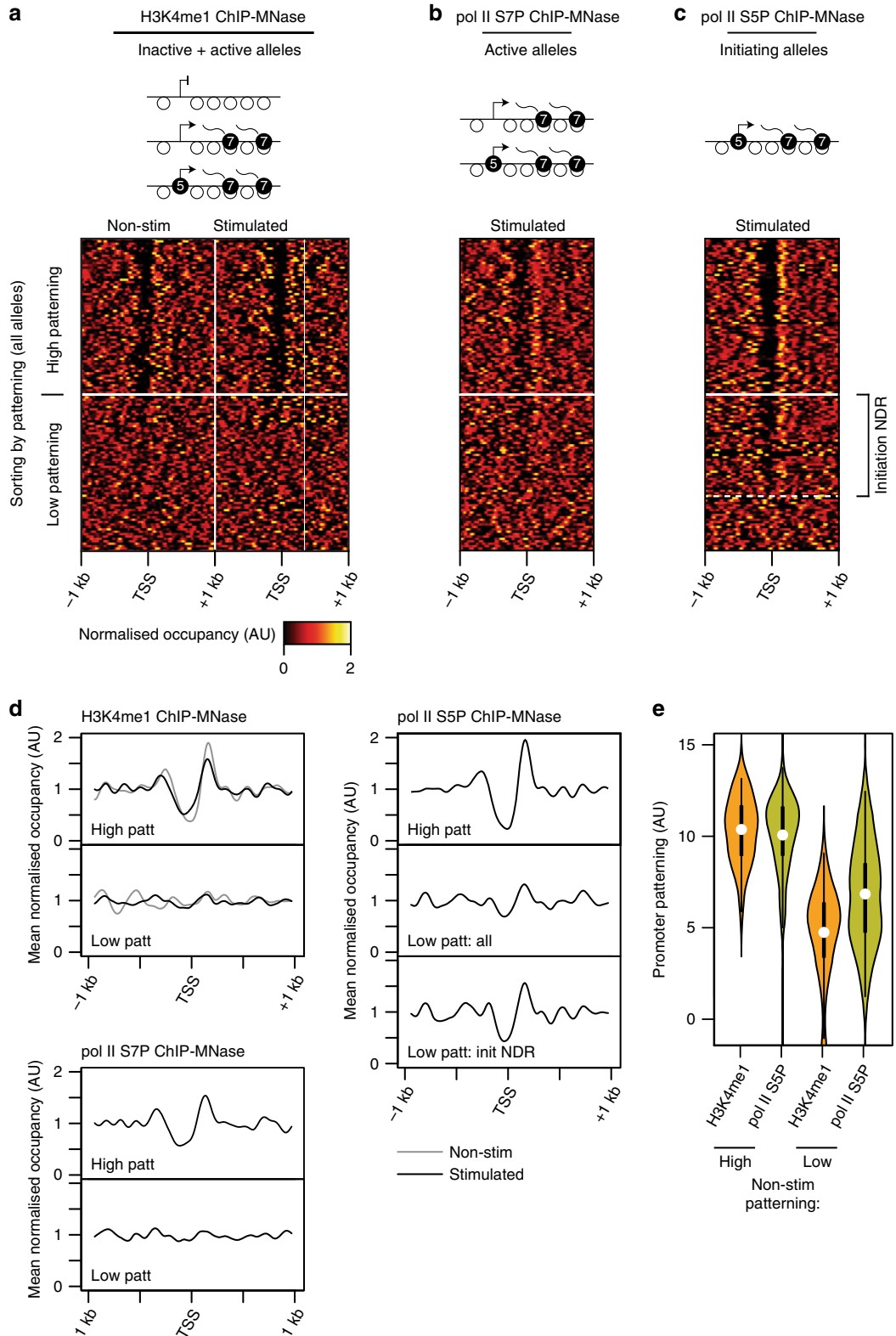

fully nucleosome-occupied TSS, it could also arise from transient nucleosome displacement during initiation, followed by rapid return of nucleosomes upon promoter clearance by RNA pol-II. To investigate this, we repeated the ChIP-MNase in stimulated fibroblasts using antibodies against S5P-modified RNA pol-II, which is present on the initiating form of RNA pol-II and typically marks the enzyme at early stages of transcription (Supplementary Fig. 16a–c[63]). Using this approach to selectively obtain nucleosome profiles of the initiating alleles (pol-II S5P) of inducible non-patterned promoters, we could reveal the appearance of a detectable NDR at the TSS of at least 62% of these promoters (Fig. 6c–e and Supplementary Fig. 16e), accompanied by a highly significant difference in quantified nucleosome patterning levels at initiating alleles compared to all alleles.

**Fig. 6 Inducible promoter activation without stable NDR formation. a** Heatmap of nucleosome occupancy levels surrounding promoters that are inducible in fibroblasts, assayed by H3K4me1 ChIP-MNase to detect both active and inactive promoter alleles in non-stimulated and TNF-α-stimulated fibroblasts. Promoters are grouped by nucleosome patterning levels in non-stimulated cells. **b** Heatmap of nucleosome occupancy levels of the same promoters as panel (**a**), assayed by RNA polII S7P ChIP-MNase to detect only active alleles, in TNF-α-stimulated fibroblasts. **c** Heatmap of nucleosome occupancy levels of the same promoters as panel (**a**), assayed by RNA polII S5P ChIP-MNase to enrich for alleles undergoing transcription initiation, in TNF-α-stimulated fibroblasts. Promoters with a clear NDR at initiating alleles but not at all inactive + active alleles are indicated ('initiation NDR'). Note that S5P marks RNA polII during initiation but also early stages of transcription elongation (see Supplementary Fig. 16a); thus, the relative enrichment of initiating over elongating alleles, which affects the detection of patterning at initiating alleles, may vary between promoters (see Supplementary Fig. 16e). **d** Profiles of mean nucleosomal occupancy assayed by H3K4me1 ChIP-MNase (top left), RNA polII S7P ChIP-MNase (bottom left) or RNA polII S5P ChIP-MNase (right), surrounding promoters with high patterning (top) or low patterning (below) in non-stimulated fibroblasts. The bottom polII S5P panel depicts the subset of promoters at which ChIP-MNase reveals a detectable NDR only at initiating alleles (indicated in panel (**c**)). Grey lines: no stimulation; solid lines: TNF-α-stimulation. **e** Violin plots of quantified promoter patterning levels at inducible promoters in TNF-α-stimulated fibroblasts, grouped by mean patterning levels at all inactive + active alleles in non-stimuated cells. Orange violins depict patterning levels at all alleles, quantified by H3K4me1 ChIP-Mnase; olive-coloured violins depict patterning levels at initiating alleles, quantified by RNA polII S5P ChIP-MNase. 82% of promoters that are unpatterned at all alleles in non-stimulated fibroblasts display patterning levels at all alleles upon stimulation that are lower then the median level at initiating alleles. Significance of difference after stimulation of patterning quantified by H3K4me1 ChIP MNase vs. RNA polII S5P ChIP MNase: high non-stim patterning: $p = 0.70$; low non-stim patterning: $p = 1.8 \times 10^{-4}$ (two-tailed Mann–Whitney $U$ test). Thick bars indicate limits of quartiles; dots indicate means.

These findings fit a model whereby nucleosome displacement or repositioning at the TSS regions of non-patterned inducible promoters is short-lived, and that these regions become rapidly re-occupied by nucleosomes between cycles of initiation. This implies that the BAF-dependent activation of these promoters is associated with transient NDR formation at initiating alleles, rather than playing a role in establishing stable alterations to promoter nucleosome patterning. Thus, the BAF complex may be repeatedly utilised to deplete nucleosomes from TSS regions to enable each cycle of transcriptional initiation by RNA pol-II.

Finally, the difference between the behaviour of non-patterned inducible promoters after activation (which display measurable patterning only at initiating alleles) from most other active promoters (which exhibit apparently stable patterning that is measurable from all alleles; Fig. 3a, c) prompted us to question whether the small fraction of non-patterned active promoters may likewise harbour NDRs that can be measured only at particular subsets of alleles. Indeed, a fraction of active promoters with the lowest levels of nucleosome patterning at all alleles nonetheless exhibit significantly higher patterning when measured at transcriptionally active (pol-II S7P) alleles, and still higher levels when measuring only initiating (pol-II S5P) alleles (Supplementary Fig. 17a–d). Such promoters could represent genes that are active in the steady state with heterogeneous cellular or allelic expresssion, or inducible genes that were activated during development or by stimuli that were not purposely controlled in our experiments. In line with the latter scenario, genes characterised by non-patterned active promoters are enriched for particular biological pathways consistent with developmental or stimulus-driven activation (Supplementary Fig. 17e). Regardless of their mode of activation, though, we estimate that around 9–10% of active promoters exhibit allele-specific or initiation-specific nucleosome patterning.

## Discussion
We have presented an approach—ChIP-MNase—to allow high-resolution analysis of nucleosome positioning at chosen genomic regions and selectively marked alleles, and used this to investigate the role of nucleosome positioning, and its regulation, at stimulus-inducible promoters. We find that the patterning of nucleosomes before stimulation can define cell-type-specific promoter responsiveness, and that inducible activation does not usually involve major changes in promoter nucleosome patterning. We also show that activation of many inducible promoters occurs without stable nucleosome depletion from the TSS, but that they require enzymatic remodelling which is associated with

transient nucleosome displacement during the initiation steps of transcription.

By enriching for genomic intervals bearing a selected biochemical property (not limited to antigens on nucleosomes), ChIP-MNase significantly increases the level of coverage across chosen features-of-interest (by 6- to 35-fold in our experiments), which is critical in order to define nucleosome positions with accuracy (Supplementary Fig. 3g). Moreover, the breadth of analysed regions is not restricted by the genomic extent of the target mark, and can be controlled by varying the level of chromatin fragmentation prior to immunprecipitation. Using replicate samples, we estimate the resolution of our inferred nucleosome positions to be around 30 bp (Fig. 1g), which is comparable to previously estimated levels of positional heterogeneity (30 bp[12]) and to measured redundant nucleosome positioning in yeast (20–40 bp[64]).

Other approaches aimed at combining nucleosome mapping with enrichment[6] or allele-specific analysis[65] have previously been reported. Hybridisation-based capture of MNase-digested DNA[51,52,66–69] can allow selection of arbitrary genomic regions to be enriched; however, it is not readily amenable to regions containing repetitive or non-unique sequences, and it cannot separate functionally distinct alleles of the same sequence. ChIP of MNase-digested chromatin has been used to enrich nucleosomes bearing defined histone modifications[15,22,32,70–75]; however, this approach is unable to directly distinguish changes in nucleosome occupancies from changes in histone modification levels[15,70,72], and it is not well-suited to map nucleosome positions surrounding non-histone targets. Finally, ATAC-seq can reveal nucleosome positions surrounding transposase-accessible genomic regions, including promoters[8,11], and affords high levels of enrichment; however, it offers no choice of enriched regions, and mapping typically only spans a few nucleosomes surrounding active elements. Thus, we feel that ChIP-MNase represents a flexible alternative approach for selective analysis of nucleosomes at defined genomic features and alleles.

A conserved phenomenon that has been observed in all eukaryotes studied so far is the characteristic patterning of nucleosomes at transcriptionally active promoters, including an NDR immediately upstream of the TSS and a strictly positioned +1 nucleosome[5–7,19]. Notably, though, whereas high-resolution nucleosome-mapping studies in yeast have shown that this is a property of essentially all active promoters, many studies in mammalian genomes have relied on analysis of averaged profiles, which does not easily allow analysis of individual promoters or small subsets. Using ChIP-MNase to analyse stimulus-inducible

promoters, we found that many inactive inducible promoters in a given cell-type are also characterised by patterned nucleosome occupancies that closely resemble those at actively transcribing promoters (Fig. 3). This agrees with a previous aggregated analysis of average nucleosome occupancies at TCR-inducible promoters in T cells[16]. However, quantitation of patterning at individual inducible promoters further revealed that high patterning levels are present at only a specific subset, and that this patterning can be cell-type dependent and predictive of promoter responsiveness to activation by stimuli (Figs. 3 and 4). Thus, establishment of nucleosome patterning appears to be associated with poising of some promoters for subsequent transcriptional activation. The presence of a pre-stimulation NDR suggests that patterned inducible promoters may be directly accessible for binding by the transcriptional machinery[16,27], and, consistent with this, we find that patterned inducible promoters are generally pre-loaded with paused RNA pol-II (Supplementary Figs. 11 and 12), and that some are associated with low-level mRNA expression even before stimulus-driven activation (Fig. 3). However, our data do not unambiguously resolve whether basal transcription and/or the presence of paused RNA pol-II represent contributory causes[16,54,56] or consequences[13,76] of promoter nucleosome patterning.

In agreement with earlier reports[13,27], we find that detection of an NDR at active and inducible promoters is favoured by the presence of a CGI. However, although CGIs represent a sequence-encoded feature that is strongly associated with nucleosome patterning[13], the existence of many inducible promoters that exhibit cell-type dependent patterning (Fig. 4) indicates that the DNA sequence is not the only determinant. Instead, our data are in line with a model whereby the presence of a CGI at inducible promoters identifies a subset that is broadly inducible or active in both cell-types analysed, whereas non-CGI-containing promoters more often correspond to those with cell-type specific inducibility or activity, and which often exhibit a cell type-dependent NDR[57].

Around half of the inducible promoters that we analysed exhibit low nucleosome patterning levels and lack a detectable NDR. Unexpectedly, this unpatterned configuration persists even after stimulus-induced activation, indicating that initiation of transcription from these promoters neither requires nor induces formation of a stable NDR at the timescale analysed. This behaviour contrasts with that of the majority of promoters that are active under steady-state conditions in the same cells, and is also markedly different from the reported behaviour of inducible stress–response genes in yeast[5,77,78]. We used the allele-specific readout of ChIP-MNase to confirm that the apparent absence of a promoter NDR is not simply a consequence of heterogeneous activation of a subpopulation of promoter alleles. Thus, the presence of a stable, pre-formed NDR is not required at these promoters to enable transcription initiation (although we cannot exclude that stable NDRs may later become established after prolonged activation[79,80]. Nevertheless, we were able to measure short-lived nucleosome depletion (or destabilization) at the TSS of many of these promoters when bound by initiating RNA pol-II (Fig. 6). Since these same promoters do not exhibit the same degree of nucleosome depletion when associated with elongating RNA pol-II, this implies that nucleosome re-assembly (or re-stabilisation) occurs between cycles of transcription initiation, and suggests that re-initiation may require most or all of the steps needed for initial activation[12].

We find that the BAF complex is generally dispensable for establishment of pre-stimulation nucleosome patterning at both active and inducible promoters. Moreover, at non-patterned inducible promoters, BAF activity is also insufficient to establish a stable NDR that persists between cycles of transcription initiation. Instead, our results support a model whereby transient nucleosome

remodelling by the BAF complex enables binding of RNA pol-II (and possibly also PIC assembly) at non-patterned promoters, and it is plausible that this active process must occur repeatedly for successive transcription initiation events. How then is the BAF complex recruited to act at promoters? Using these publicly available data, we found that the BAF complex binds to most inducible promoters in non-stimulated cells, before promoter activation. Thus, it seems likely that the BAF complex associates with chromatin features or factors that are already present at inducible promoters in the inactive state, further supporting the notion that the responsiveness of these promoters is preconfigured before stimulation.

The set of genes that require the remodelling activity of the BAF complex for full activity in fibroblasts is rather limited, comprising only 8% of expressed genes in this cell type. However, it is enriched among genes with cell type-specific expression, and strongly enriched among stimulus-inducible genes. This concurs with its previously reported role in developmentally regulated and inflammatory gene expression[24,27,61,81]. Our finding that BAF-dependent genes are typically characterised by unpatterned promoters, which lack a stable NDR, suggests that aspects of promoters that favour efficient nucleosome assembly may be exploited as a strategy to achieve highly regulated expression. In line with this, sequences favouring high nucleosome occupancy are frequently selected at promoters and gene regulatory regions[82,83].

In summary, in this report we have used the ChIP-MNase approach to characterise nucleosomal behaviour at high-resolution during activation of stimulus-inducible promoters. We find that the patterning of nucleosomes at individual promoters can vary substantially between cell-types, but that it is strikingly constant during stimulus-driven activation, and transcription initiation is linked to mainly transient remodelling events. Altogether, our results show that nucleosome patterning can define both the cofactor requirements and cell type-specific responsiveness of inducible promoters.

## Methods

**Cell culture and stimulation.** Progenitor cell lines derived from mouse bone marrow were cultured under non-differentiating conditions in 20 ng ml⁻¹ SCF, IL-3 (0.1% X63-IL3 supernatant) + IL-6 (2% X63-IL6 supernatant). DCs were differentiated from progenitor cells in granulocyte-macrophage colony-stimulating factor (GM-CSF) (4% X63-GMCSF supernatant) + 5 ng ml⁻¹ SCF for 4–5 days, followed by GM-CSF alone for 4–5 days[84]. DCs were adherent and >90% MHC class II + CD11c+ by flow cytometric analysis. DCs were stimulated for 1 h with 100 ng ml⁻¹ LPS. 3T3 fibroblasts were stimulated for 1 h with 5 ng ml⁻¹ TNF-α.

**ChIP-MNase sample preparation and sequencing.** Cells were fixed by addition of formaldehyde at a final concentration of 4% for 10 min at room temperature. Fixed cells were washed once quickly and twice for 10 min each with ice-cold phosphate-buffered saline (PBS), and collected by scraping and centrifugation for 10 min at 500g at 4 °C. Cells were resuspended at $2 \times 10^7$ ml⁻¹ and nuclei were released by incubation in ice-cold L1 buffer (50 mM Tris pH8, 2 mM EDTA, 0.1% NP40, 10% glycerol + protease inhibitors) for 5 min. Nuclei were collected by centrifugation for 5 min at 500g at 4 °C and washed twice in ice-cold buffer D (50 mM Tris pH8, 5 mM magnesium acetate, 0.1 mM EDTA, 5 mM DTT, 25% glycerol + protease inhibitors). Washed nuclei were resuspended in MNase buffer (10 mM Tris pH7.4, 15 mM NaCl, 60 mM KCl, 0.25 M sucrose, 0.5 mM DTT, 1 mM CaCl₂) at $2 \times 10^7$ cells in 400 μl⁻¹ and lysed by two freeze–thaw cycles and sonication (4 cycles of 10 s sonication, 30 s recovery on ice, using a micro-tip sonicator). For each sample, six 20 μl aliquots (containing $10^6$ lysed nuclei) were removed and digested with titrated amounts of MNase (5-fold dilutions from 4 U to 0.0013 U MNase per $10^6$ lysed nuclei) at 25 °C for 30 min with shaking, reactions were stopped by addition of EDTA to a final concentration of 12.5 mM, fixation was reverted by addition of an an equal volume of 2% sodium dodecyl sulfate (SDS) solution and overnight incubation at 65 °C, samples were diluted to 500 μl with PBS and digested with 100 μg ml⁻¹ proteinase K for 1 h at 55 °C, and DNA was purified using 300 μl of Miniprep Express matrix. Each digested aliquot was analysed on an agarose gel to determine the dose of MNase required to generate the desired fragment size for ChIP (typically 5–10 kb in this study). This dose was used to digest the remainder of each sample in several 200 μl reactions (containing $10^7$

lysed nuclei). After stopping reactions, debris was pelleted by centrifugation and supernatants containing chromatin were pooled and used for ChIP. Four parallel immunoprecipitations of chromatin from $2 \times 10^7$ nuclei were performed for each sample. 1 ml of buffer V (50 mM Tris pH7.4, 50 mM NaCl, 5 mM EDTA) was added to each 400 µl of digested chromatin, and samples were pre-cleared using 40 µl of protein-G sepharose for 1 h at 4 °C with rotation. After centrifugation at 500$g$ for 1 min, 4 µg of antibody was added to the supernatant containing the pre-cleared samples and incubated overnight at 4 °C with rotation. Antibody-bound chromatin was pulled-down by addition of 15 µl of protein-G sepharose for 30 min at 4 °C, and collected by centrifugation. Chromatin-bound beads were washed once quickly and 3 times for 5 min each with 1 ml of ice-cold buffer WBNS (20 mM Tris pH8, 500 mM NaCl, 2 mM EDTA, 1% NP40), followed by two washes for 5 min each with 1 ml of ice-cold MNase buffer. A 50 µl aliquot of each sample was collected at the penultimate wash step for verification of ChIP efficiency. Immunoprecipitated chromatin was resuspended in MNase buffer to a final volume of 30 µl (including 15 µl of beads), and digested using titrated amounts of MNase (2.5, 1.25 and 0.25 U MNase per 30 µl reaction containing immunoprecipitated chromatin from $2 \times 10^7$ nuclei) at 25 °C for 30 min. Each reaction was stopped by addition of EDTA to a final concentration of 12.5 mM, fixation was reverted by addition of 70 µl of 2% SDS solution and overnight incubation at 65 °C, and DNA was directly purified using the MinElute PCR purification kit (Qiagen), and eluted in 30 µl elution buffer. DNA samples were quantified using picogreen reagent and a Qubit fluorometer (Invitrogen). ChIP efficiency was verified by quantitative polymerase chain reaction (qPCR) using pre-digestion samples, with primers specific for known positive and negative control regions according to the antibody target. MNase digestion efficiency was assessed by anaysis of 1 µl of each digestion using Agilent HS DNA chips, and for each initial sample, the digestion yielding approximately 80% mononucleosome-sized fragments was selected for sequencing. This was chosen to encourage complete digestion of inter-nucleosomal DNA, while limiting the scope for MNase nibbling into the edges of nucleosome-bound regions (which is revealed by the appearance of sub-nucleosomal-sized DNA fragments in over-digested samples; see 50 U digestion in Supplementary Fig. 1a). For single-end sequencing, mononucleosome-sized fragments in the range 120–170 bp were gel-purified; for paired-end sequencing, all DNA fragments were used, and mononucleosome-sized fragments were identified during data analysis. Sequencing libraries were prepared using the NebNext Ultra II DNA library kit (New England Biolabs), and sequenced using HiSeq2000 or NextSeq500 instruments (Illumina).

**Microarray gene expression analysis.** Cells were lysed directly into RNA lysis buffer (38% phenol, 0.8 M guanidine thiocyanate, 0.4 M ammonium thiocyanate, 0.1 M Na acetate pH5, 5% glycerol[85]. Total RNA was separated by addition of 0.2 volumes of chloroform, emulsification by shaking, and centrifugation at 13,000$g$ for 10 min at 4 °C, followed by precipitation of RNA in the upper, aqueous phase by addition of 1 µl of glycogen azure (Sigma) plus 2.5 volumes of isopropanol, centrifugation at 13,000$g$ for 20 min at 4 °C, and washing the RNA pellet in 70% ethanol. RNA samples were dissolved in 20 µl water, quantified using a Nanodrop spectrophotometer (Thermo), and subsequently processed for microarray analysis using Affymetrix GeneChip mouse gene 1.0 ST arrays. Three independent biological replicates were performed on different days and analysed for each experimental group.

**ChIP sequencing.** Cells were washed twice with PBS at room temperature, and fixed by addition of 20 µl of 0.5 M DSG to cells in 5 ml PBS and incubation for 45 min at room temperature. Cells were washed again twice with PBS at room temperature, and double-fixed by addition of formaldehyde to a final concentration of 1% to cells in 5 ml PBS and incubation for 15 min at room temperature. Fixation was stopped by addition of glycine to a final concentration of 125 mM. Fixed cells were washed once quickly and twice for 10 min each with ice-cold PBS, and collected by scraping and centrifugation for 10 min at 500$g$ at 4 °C. Cells were resuspended at approximately $1 \times 10^7$ ml$^{-1}$ and nuclei released by incubation in 1 ml of ice-cold L1 buffer (50 mM Tris pH8, 2 mM EDTA, 0.1% NP40, 10% glycerol + protease inhibitors) for 5 min. Nuclei were collected by centrifugation for 5 min at 500$g$ at 4 °C and resuspended at $5 \times 10^7$ ml$^{-1}$ in 900 µl of ice-cold L2 buffer (50 mM Tris pH8, 5 mM EDTA, 1% SDS + protease inhibitors). Chromatin was fragmented by sonication to an average size of 600–700 bp (typically 9 cycles of 10 s sonication, 1 min recovery on ice, using a micro-tip sonicator) and insoluble debris was pelleted by centrifugation. A 50 µl aliquot was removed from each sample and analysed by agarose gel electrophoresis after DNA extraction to verify fragmentation. Fragmented chromatin was diluted with 9 volumes of buffer DB (50 mM Tris pH8, 200 mM NaCl, 5 mM EDTA, 0.5% NP40) and pre-cleared using 40 µl of protein-G (for mouse monoclonal antibodies) or protein-A (for rabbit polyclonal antibodies) sepharose for 1 h at 4 °C with rotation. After centrifugation at 500$g$ for 1 min, 4 µg of antibody was added to each 1 ml of supernatant containing the pre-cleared samples and incubated overnight at 4 °C with rotation. Antibody-bound chromatin was pulled-down by addition of 15 µl of protein-A or -G sepharose for 30 min at 4 °C, and collected by centrifugation. Chromatin-bound beads were washed once quickly and 4 times for 5 min each with 1 ml of ice-cold buffer WB (20 mM Tris pH8, 500 mM NaCl, 2 mM EDTA, 1% NP40, 0.1% SDS), followed by 3 washes for 5 min each with 1 ml of ice-cold TE. Immunoprecipitated chromatin was released by incubating beads in buffer EB (TE + 2% SDS) for 5 min

at room temperature with periodic tickling, and the supernatant was collected. This was repeated two more times and the supernatantw were pooled. Fixation was reverted by overnight incubation at 65 °C, and DNA was directly purified using the MinElute PCR purification kit (Qiagen), and eluted in 30 µl elution buffer. DNA samples were quantified using picogreen reagent and a Qubit fluorometer (Invitrogen). ChIP efficiency was verified by qPCR with primers specific for known positive and negative control regions according to the antibody target. Sequencing libraries were prepared using the NebNext Ultra II DNA library kit (New England Biolabs), and sequenced using HiSeq2000 or NextSeq500 instruments (Illumina).

**Stable shRNA knock-down.** $Brg1$ and $Brm$ mRNAs were stably knocked-down using an shRNA with sequence <u>tggagaagcagcagaagatt</u>*TCAAGAG*aatcttctgctgcttctcca (target sequence underlined; loop sequence in italics[24]; expressed from the mouse U6 promoter in the retroviral vector pSirΔ-U6CG, which drives co-expression of green fluorescent protein (GFP). Fibroblasts were retrovirally transduced with supernatants from transfected ecotropic Phoenix packaging cells, and sorted for high GFP expression. The level of residual $Brg1$ and $Brm$ mRNA (Fig. 6a) was measured by quantitative RT-PCR, using primers agagaagcagtggctcaagg and agatttcttctgccggacct (to detect $Brg1$) and gccagtggatttcaaaaagataaa & ttgtgacagagaagcatgacg (to detect $Brm$).

**Antibodies.** Monoclonal anti-HeK4me1 (clone CMA-302) was provided by H. Kimura[86]. Anti-pol-II S7P (clone 4E12; cat 04-1570) was from Millipore, anti-pol-II S5P (cat ab5121) was from Abcam, anti-p65 (C20, cat sc732) was from Santa Cruz biotechnology.

**ChIP-MNase data processing.** Demultiplexed sequence data was aligned to the mouse mm9 genome using bowtie (with options -v 2 -a -m 5 --tryhard for single-end sequencing, or -v 2 -a -m 5 --maxins 2000 --tryhard for paired-end sequencing); alignment rates were typically around 80% for H3K4me1 ChIP-MNase, and around 90% for pol-II ChIP-MNase. Excess duplicate reads that mapped to the exact same genomic location (corresponding to likely PCR duplicates) were eliminated if there were significantly more than expected based on the local read density in a range of ±2 bp at a $p$ value of 0.05; the level of redundant reads that were filtered-out at this stage was typically around 10%. For paired-end sequencing, only mononucleosome-sized fragments in the range 140–180 bp were retained for subsequent analysis; for single-end sequencing, fragments were estimated by extending reads to the average fragment size derived from the distribution of distances between nearby reads that mapped to the genome in opposite orientations. Fragments mapping within 10 kb of refGene annotated promoters were selected, and only fragments that were uniquely mapped within this set were retained (in other words, fragments that mapped uniquely among promoter regions were allowed even if they aligned to other non-promoter genomic regions, since we considered that the promoter mapping was most likely to be the true origin of the fragment due to the ChIP enrichment strategy used). The total number of fragments mapping within bins spanning 10 bp genomic intervals were counted, and these counts were normalised according to the total number of fragments mapping within the surrounding 1 kb genomic interval. Nucleosome occupancy was calculated as the mean number of normalised counts in overlapping 50 bp bins centred every 10 bp, and the levels were calculated for every position from −1 kb to +1 kb surrounding refGene annotated promoters.

**Analysis of ChIP-MNase enrichment at promoters.** The mean number of fragments per bp in each dataset mapping between −10 kb and +10 kb of each annotated refGene promoter was counted, and the fold-enrichment at each promoter was calculated by dividing this number by the mean genome-wide density of mapped fragments in the dataset (that is, the number of mapped fragments divided by the mappable genome size). High coverage promoters, which were used in subsequent analyses, were defined as those with a mean coverage of ≥0.05 fragments per bp in the range −1 kb to +1 kb in non-stimulated DC and fibroblast datasets (equivalent to 100 mapped fragments within the 2 kb range, or approximately 4-fold enrichment at the sequencing depths used in this study), and amounted to 13,559 promoters, or 53% of the total of 25,881 refGene annotated promoters. To ensure that this criterion did not result in exclusion of promoters of non-expressed or low expressed genes, the fraction of promoters of genes with distinct expression levels was calculated among these high coverage promoters, or among the 10% of promoters with the highest fold-enrichment (shown in Fig. 1c). This confirmed that promoters of non-expressed genes are not excluded, although they are detectably under-represented (by 0.46- or 0.48-fold, respectively). Promoters of expressed genes are over-represented at all expression levels. The same approach was used to calculate the fold-enrichment at CGI-promoters and promoters of inducible genes (shown in Supplementary Fig. 2c–e).

**Estimation of nucleosome positions and resolution.** To predict preferred nucleosome positions, all local maxima (peaks) were first identified throughout the 10 bp-resolution occupancy data. Next, to distinguish discrete peaks (likely reflecting true preferred nucleosome positions) from low-level measurement noise, we calculated the topographic prominence of each peak, defined as the height of each peak above the lowest valley separating it from an adjacent higher peak

(Supplementary Fig. 3c). The distribution of prominences is approximately linear across the range of peak heights, but shows an overrepresentation at very low levels. Peaks with these very low prominences were excluded by applying a cut-off (excluding roughly 20% of all peaks; Supplementary Fig. 3d); this relaxed criterion is intended to encompass moderately preferred nucleosome positions, and to allow the possibility of multiple preferred nucleosome positions at any genomic nucleosomal interval. Note that more-stringent cut-offs select higher-confidence nucleosome positions and display a strongly phased arrangement of inter-nucleosomal spacing, but may overlook potentially valid but less-favoured positions (Supplementary Fig. 3e). To estimate the resolution, distance discrepancies were determined between each predicted nucleosome position and the closest position that was independently predicted from a replicate biological sample: the data resolution is defined as the mean of all discrepancies. An identical workflow was applied to genomic DNA fragmented by sonication to determine the upper limit that can be empirically measured by this approach (which is close to the resolution expected by random placement of regions at the same density; Fig. 1g, Supplementary Fig. 3f). To determine the contribution of biological or methodological variability to the estimated resolution, sequence read data from replicate samples were combined and randomly reassorted into two mock replicates (thereby eliminating any underlying differences between samples), and the same workflow was applied to the in silico reassorted mock replicate datasets. To estimate the resolution corresponding to distinct levels of sequence coverage, genomic 4 kb intervals were assigned to groups defined by coverage level (within a twofold range for each group), and the resolution and mean coverage level was determined separately for every group.

**Hierarchical clustering of samples**. Reproducibility between samples was assessed by hierarchical clustering of normalised nucleosome occupancy levels, in 10 bp bins surrounding each TSS ±1 kb. Mean occupancy profiles were directly clustered based on the simple Euclidian distance between the square-root of mean levels at each relative position in each sample. To cluster individual occupancies, the distances between the occupancy levels across every promoter region in each sample and its counterpart in every other sample were calculated in the same way and averaged, to define an overall mean distance between each pair of samples.

**Nucleosomal period and footprints**. To calculate the mean inter-nucleosomal period, the distance between every mapped fragment to all other fragments on the each strand within a 1 kb range was calculated, and a discrete Fourier transform was applied to the distributions of the number of occurrences of all distances on each strand separately: the period with highest magnitude signal was estimated by interpolation, and the offset in bp was calculated from the difference in phase at that period between the transformations of same-strand and opposite-strand distributions. The nucleosomal footprint was defined as the difference between the period and the offset calculated in this way. Estimation of the mean distance between peaks using a linear fit to the distribution of same-strand distances (to determine the period) and of the major peak of opposite-strand distances (to determine the nucleosomal footprint) gave comparable results (see Supplementary Fig. 4a, b, d).

**Analysis of nucleosome gaps at DNase hypersensitive sites**. DNase hypersensitive sites (DHS) analysed in fibroblasts correspond to ENCODE narrowPeak annotations (GEO accession number GSM1014177). To exclude promoters, only DHS midpoints >1 kb from annotated TSSs were analysed. The midpoint of each DHS was defined as the median cut site position, that is, the position at which half of all sequence tags mapped within the DHS region are mapped on either side. Normalised nucleosome occupancies were aligned relative to this position. To identify predicted single-nucleosome-sized gaps within the ChIP-MNase data, we calculated the distances between all adjacent predicted nucleosome positions, and selected those in the range of 330–370 bp from each replicate fibroblast dataset. We considered only those gaps that were independently identified in both replicates, and that fell within 1 kb intervals with a minimum coverage of 0.025 fragments per bp (25 mapped fragments within the 1 kb range). DNase hypersensitivity data was aligned relative to the midpoint of each gap. Gaps were deemed to correspond to DHS regions if the mean density of DNase sequence tags across the 400 bp interval centred on the nucleosome gap exceeded 0.1 per bp (40 mapped tags per gap, corresponding to the highest 0.5% of genome-wide DNase sensitivity levels). Note that although 39.5% of predicted gaps were classed as DHS using this criterion, this represents a lower bound for the true fraction of gaps that are DHS, since it is limited by any inaccuracies in the predition of nucleosome positions.

**Classification of promoters**. Active promoters were defined as those of genes with expression levels in non-stimulated cells (measured by RNA-seq) greater than 0.02× the mean expression level of all genes, and excluding inducible promoters. Inactive promoters were defined as those of genes with no detectable expression in non-stimulated cells (RNA-seq FPKM equal to zero) and excluding inducible promoters. Inducible promoters were defined as those of genes that exhibited a stimulus-driven increase in expression (measured by Affymetrix microarray RMA fluorescence differences) greater than 1 (for LPS-stimulated DCs) or greater than 0.35 (for TNF-α-stimulated fibroblasts), and for which the increase is statistically

significant at an α-level 0.05 (by Student's $t$ test of replicate samples). CGI promoters were defined as those with a GC content of >50% and observed/expected ratio of CG-dinucleotides of >0.6 across the region spanning the TSS ± 200 bp[87]. BAF-dependent and -inhibited promoters were defined as those of genes that exhibited decreased or increased expression, respectively, in BAF-knockdown cells greater than 0.3 (measured by Affymetrix microarray RMA fluorescence differences in replicate samples).

**Hierarchical clustering of promoter classes**. The relative similarity between classes of promoters was assessed by hierarchical clustering of normalised nucleosome occupancy levels, in 10 bp bins surrounding each TSS ± 1 kb. Mean occupancy profiles were directly clustered based on the simple Euclidian distance between the square-root of mean levels at each relative position in each sample. To cluster promoter classses based on occupancies at individual promoters, an equal-sized random sample of promoters was drawn from each class, and pairwise distances between the occupancy levels across each promoter region in each class were calculated in the same way from every other promoter in every other class, and the overall mean distance between each pair of classes was defined as the mean of all pairwise promoter distances. Clustering was also performed by including only active promoters with expression levels greater than the mean expression level of all genes, and excluding all inducible promoters with basal expression greater than 0.5× the mean expression level of all genes (Supplementary Fig. 7gi) or excluding all inducible promoters with non-zero gene expression (Supplementary Fig. 7gii); both of these analyses gave similar results, confirming that the observed clustering of inducible with active promoters cannot be driven by a subset of inducible promoters with the pre-stimulation expression levels that overlap those of active promoters, or even with any measurable level of expression.

**Quantitation of promoter patterning**. Nucleosomal patterning at individual promoters was quantified by comparing the normalised occupancy profile at each promoter to the profile of mean normalised occupancy at all active promoters, surrounding each TSS ± 1 kb. The patterning level was calculated from the sum of the products of each normalised profile after subtracting the mean occupancy level, in 10 bp bins (Supplementary Fig. 5a–c). Inducible promoters were grouped into high- and low-patterned subsets based on their patterning levels relative to the median level determined at all inducible promoters in the same cell type.

**Hierarchical clustering of individual inducible promoters**. Inducible promoters were clustered according to their level of patterning calculated using the square root of occupancy levels at each promoter position (this prevented disruption by aberrantly high occupancy levels at isolated promoter positions, and maintained consistency with the approaches used for clustering nucleosome occupancies at distinct promoter classes and cell types). Hierarchical clustering was performed using the Euclidian distances between levels, and the Ward's minimum variance method to merge clusters.

**ChIP-seq data processing**. Demultiplexed sequence data was aligned to the mouse mm9 genome using bowtie (with -v 2 -a -m 5 --maxins 2000 --tryhard for paired-end sequencing); alignment rates were around 78%. Excess duplicate reads that mapped to the exact same genomic location (corresponding to likely PCR duplicates) were eliminated if there were significantly more than expected based on the local read density in a range of ±2 bp at a $p$ value of 0.05; the level of redundant reads that were filtered out at this stage was around 6%. The total number of fragments mapping to each genomic interval were counted; fragments that mapped ambiguously to more than one genomic location were counted fractionally at all aligning locations.

**Determination of polymerase pause indices**. The levels of RNA pol-II S7P or of unmodified RNA pol-II (using dataset GSM1624320) at promoters and within gene bodies were calculated as the mean ChIP coverage levels across the TSS ± 500 bp (1 kb interval, for promoters) or across the region between +1 and +6 kb (5 kb interval, for gene bodies). The pause index for each promoter was defined as the ratio of pol-II levels at the promoter to the levels at the gene body.

**Public datasets used**. Gene and promoter annotations are from refGene (refGene table downloaded 08112013). Datasets used to analyse gene expression in DCs (GSM624282), gene expression in fibroblasts (GSM970853), gene activation in LPS-stimulated DCs (GSM799198-9, 201-2, 204-5; series GSE32255[84]), gene activation in TNF-α-stimulated fibroblasts (GSM318681-6; series GSE12697[88]), DHSs in fibroblasts (GSM1014177), MNase-seq in fibroblasts (GSM2538320-2538343; series GSE96688), BAF-155 ChIP-seq (GSM1835955)[89–93] and Brg1 ChIP-seq (GSM1835956 and GSM1132963) were retrieved from GEO.

**Reporting summary**. Further information on research design is available in the Nature Research Reporting Summary linked to this article.

## Data availability

The data that support this study are available from the corresponding author upon reasonable request. All datasets generated in this study are available from the NCBI Gene Expression Omnibus (GEO) database with accession number GSE142170. The source data underlying Figs. 1g, 2c, d, and 5a, and Supplementary Figs. 2c, 3g and 9d, are provided as a source data file.

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

## Acknowledgements

We are grateful to the teams of Ulrike Bönisch (MPI-IE), Vladimir Benes (EMBL Gene-Core), and the IRCAN genomics core facility (supported by FEDER, Région PACA, Conseil Départemental 06, ITMO Cancer Aviesan and INSERM) for sequencing, the MPI-IE and IRCAN bioinformatics services for computing resources, Hiroshi Kimura for providing anti-H3K4me1 monoclonal antibodies and Ritwick Sawarkar for critical reading of the paper. This work was supported by funding from the Max Planck Society, the German Research Foundation, the Ligue Contre le Cancer, and the Agence Nationale de la Recherche. SS is supported by INSERM.

## Author contributions

The project was conceived by all authors together; A.O. developed the ChIP-MNase technique; A.O. and S.S. performed experiments and analysed data; D.v.E. performed bioinformatic analysis; S.S. and D.v.E acquired funding and supervised the project; D.v.E. wrote the original paper, which was reviewed and edited by S.S.

## Competing interests

The authors declare no competing interests.
