## [Peer Review File · Nature Communications]

Reviewers' comments:

Reviewer #1 (Remarks to the Author):

This is a clear and well written manuscript from the van Essen group. The van Essen group has a strong background in relating immunobiology (NFkB response) to chromatin structure (primarily HPTMs). In the current manuscript they ask the question: What is the role of nucleosome position at cell type specific inducible promoters? To answer this question they generate high resolution nucleosome position maps by first ChIP-capturing promoter regions (as defined by H3K4me1) followed by MNase digestion to generate nucleosome distribution maps. They use these nucleosome maps to demonstrate that nucleosome organization at inactive inducible promoters may predict cell-type specific response.

COMMENTS:

Is ChIP_MNase approach comprehensive in its ability to pull down promoter regions? In figure 1.B.ii where the "Proportion of promoters of non-expressed and expressed genes in fibroblasts among all promoters (left) or among those with the 10% highest levels of coverage after enrichment by H3K4me1 ChIP-MNase." It is important here to know what the total numbers of promoters pulled down is. Another way to answer this is what is the total number of rows in 3.A.i.

What is the actual enrich at promoter regions? The authors state that "the breadth of analyzed regions is not restricted to the genomic extent of the target mark since it is defined by the initial chromatin fragmentation." It would be reassuring to see that the enrichment works well by calculating the the enrichment at promoters not solely in terms of Fig 1.B.i, but as a calculation of % of Paired end reads that fall within 5KB of a TSS. If the goal of the ChIP enrichment is to reduce the sequence space to generate higher resolution maps then a majority of reads should map to <10% of the genome (or some quantitative global measure of coverage of promoters across the genome).

Regarding the "Correction of ChIP-MNase signal for large-scale variation in the magnitude of ChIP recovery." Some acknowledgement should be made that the tradeoff for this type of correction would be to make nucleosome position signal comparable across a diverse set of promoters, but at the loss of occupancy information (see PMID 20683475). This is not a criticism of the approach, rather a clarification of what is being measured. In fact this normalization of occupancy information strengthens the use of the word "position" in the title.

Regarding "we find that up to 40% of these are marked by elevated DNase sensitivity (figure 2d)" Does this result suggest that 60% of the promoters pulled down in the ChIP-MNase experiment lack a NDR at the TSS? Figure 3.A.i suggests otherwise. or is it that the NDR is not always identified as a DHS?

The prose regarding the approach in making Figure 4 is terribly dense and more challenging to understand than it needs to be. Please consider rewriting.

Figure 4.a suggests that a large proportion of active genes are "non-patterned." As nucleosome position patterning is used as a measure of activity and a predictor of responsiveness (because they look like active genes), the unpatterned active genes should be discussed a little bit. Perhaps they are a specific ontologic category? Additionally, are these unpatterned active genes included in the downstream analysis?

"However, BAF-dependent activation of non-patterned inducible promoters is not accompanied by establishment of a stable NDR or conversion to a patterned nucleosome arrangement resembling other active promoters (figures 5d,e, S5d), and the absence of a stable NDR at non-patterned inducible promoters is unchanged in BAF-knockdown cells (figures 5d,e, S5d)." Given that the

remodeling event is temporary, as stated in the next section of the paper, could another likely interpretation of these results be that an NDR is established, but the temporal resolution of the experiment does not allow the observation of the temporary NDR formation (see PMID: 24310001, 26771136).

This reviewer simply does not understand the use of "allele specific" throughout this manuscript, and the language should be tightened up. The expectation set up in the second sentence of the summary is that differential organization at the maternal and paternal alleles of the same gene will be resolved using SNPs from the individual sequencing reads. This is not the case, the use of "allele" throughout this manuscript is confusing.

Reviewer #2 (Remarks to the Author):

Oruba et al. present a new method ChIP-MNase. The allure of this method is that it gives an alternative to MNase-seq for large mammalian genomes, where getting good coverage of nucleosomes genome-wide is prohibitively expensive. Their method involves pulling down specific chromatin constituents (H3K4me1 for example) that is part of ~10 kb fragments and then further digestion to mono-nucleosomes of the pull down material. Thus, they are supposed to get a ~10 kb view of nucleosomes around epitope of interest. Using this technique, they seek to map promoter nucleosomes at high-resolution with minimal sequencing, in ground state and stimulated dendritic cells and fibroblasts. First, they observe nucleosome depletion at the promoter and nucleosome patterning akin to active genes at inducible genes even in the absence of stimulation in dendritic cells. Furthermore, they determine the status of inducible genes of dendritic cells in fibroblasts. They observe a set of these genes are active, another set inducible and the rest inactive, as would be expected. Furthermore, the nucleosome depletion status and/or the nucleosome pattern in the gene body is not conserved between the cell types, which strongly suggests that cell type-specific factors are determining nucleosome landscape in the inducible genes (prior to stimulation also). They then tried to deplete BAF using shRNA and observed minimal effect of this knockdown on expressed genes and the inducible genes that had NDRs prior to stimulation. The inducible genes with occluded promoters are affected the most due to BAF knockdown. Finally, they demonstrate ChIP-MNase with Ser5 and Ser7 phosphorylated RNA polymerase II (RNAPII). Here, they find a striking phenomenon where, inducible genes with occluded promoters as observed by H3K4me1 ChIP-MNase, had nucleosome depletion when profiled by initiating RNAPII (Ser5 phosphorylated), indicating that there is transient remodeling of these promoters during initiation.

Overall, this is a useful method to save sequencing costs, but the presentation needs to be condensed and improved (suggestions below) for a wide audience to appreciate this study. Furthermore, the conclusions regarding determinants of nucleosome depletion at inducible genes can be strengthened based on both prior research and possible analysis with their own datasets.

Specific comments:

1. Page 4: "The requirement for much lower sequencing depths to achieve high coverage across regions-of-interest using ChIP-MNase facilitates the inclusion and analysis of replicate biological samples."

Page 4: "substantially lower levels of sequencing than are required by non-targeted approaches."

The Authors need to present some data regarding the reduction in total sequencing reads possible by ChIP-MNase compared to global MNase. Since this is the main advantage of the method presented here, subjective statements (quoted above) in the text are not enough to prove the utility of the method. Overall, I agree with the Authors that this would be cheaper, but this needs to be presented explicitly and quantitatively.

2. For all the following quotes/figures, just the mean values are presented, which preclude readers from evaluating the quality and/or biology revealed by the data. These should be replaced by histograms or violin plots that show the full distributions:

Page 3: "The increased coverage of fragments mapping to ChIP-enriched regions compared to other genomic locations, which serves as an estimate of the specificity of the measured signal for alleles bearing the immunoprecipitated mark, was typically around 30-fold (figure S1i)"

Page 4: "The mean difference in nucleosome positions predicted using ChIP-MNase datasets generated from individual replicate samples is approximately 30bp"

Page 4: "we measured mean spacings of 192bp in fibroblasts and 183bp in DCs by ChIP-MNase (figure 2a; note that the precision attained by averaging data from millions of mapped fragments can greatly surpass the 30bp resolution of individual nucleosome positions)."

3. Page 5: "We find that the 'phased' pattern of mean nucleosome occupancy is largely undiminished both downstream and upstream of the TSS when nucleosomes are individually aligned (figure S2l,m)"

Here, I assume 0 represents the position of the centered nucleosome. The y-axis is cut-off so that we can't even see the centered nucleosome. The adjacent nucleosomes have signal around the mean. This would mean that the probability of observing adjacent nucleosomes is really low, undermining the claim of the Authors that the MNase-ChIP has alleviated the read depth issue of MNase-seq.

4. Page 6: "Strikingly, we observed that nucleosome patterning at many - but not all - inducible promoters closely resembles that at active promoters, even in non-stimulated cells before induction of gene expression."

Discussion: "Similarly, we cannot unambiguously determine from our data whether basal transcription represents a cause or consequence of nucleosome patterning at some promoters."

The fact that most of these genes have paused Pol2 has to be stated when this statement is made. Furthermore, presence of paused Pol2 at inducible genes indicates that stimulation might regulate elongation. This is the simplest explanation for presence of NDRs at many inducible genes. The Authors could examine RNAPII-Ser5p data in un-stimulated state to confirm this.

5. Page 7: "nor does it arise through preferential recovery of a minority of patterned alleles by the H3K4me1 ChIP-MNase technique (figure S3h-k)."

The Authors should separate patterned induced and non-patterned induced genes to confirm to confirm that they see same feature by MNase also. This would rule out the possibility that they are observing effect of pulling down H3K4me1.

6. Page 10: "Therefore, establishment and maintenance of a stable NDR at the promoters of most expressed genes does not require the normal remodelling activity of the BAF complex."

The Authors do not show BAF complex protein levels after knockdown. The perdurance of BAF remodelers after knockdown by RNAi could explain the results they observe. One possibility is that BAF complex is knocked down only marginally at the protein level, and the levels are enough for continued function at active genes and inducible genes with NDRs. Whereas, the levels are not enough for BAF function at inducible genes without NDR.

7. A similar approach was described in this paper:

<https://www.ncbi.nlm.nih.gov/pubmed/31371436>, which needs to be discussed.

Reviewer #3 (Remarks to the Author):

First of all, I would like to emphasize that this is a very thorough and well-described study of nucleosome positioning in mammalian cells. The authors correctly point out that previous studies of this type have been somewhat hindered by low sequencing depths and the absence of replicates. The authors circumvent this difficulty by employing what they call ChIP-MNase, with the H3K4me1 histone modification as the ChIP target. Thus they trade genome-wide coverage for a higher sequencing depth, at the risk of creating a biased sample (the authors argue that H3K4me1 is safe in this regard because it is found at both active and inactive promoters and I would tend to agree with this assertion).

Although the ChIP-MNase technique is described as novel in the manuscript, it is clearly a combination of well-known methods for exploring chromatin structure. Similar protocols have been developed in Struhl, Rando, Broach labs and probably many others.

The authors carry out a thorough analysis of nucleosome maps under various conditions. Their main findings include:

1. Observation of different internucleosome linker lengths in 3T3 vs. DC cells.
2. Nucleosome "patterning" or periodicity is much more pronounced in active vs. inactive promoters, with the lack of patterning in the latter attributed to the lack of phasing between periodic nucleosome arrays in different inactive genes.
3. Observation of nucleosome depletion at DNase hypersensitive sites.
4. Stimulus-inducible promoters sometimes display patterning of active promoters even before stimulus-driven activation (ie, they are "primed" to be induced, potentially through pre-loading of RNA PolII before stimulation).
5. Nucleosome positioning at many inducible promoters is not dictated by DNA sequence alone (this is already evident from the significantly different nucleosome spacing in 3T3 vs. DC cells).
6. Using BAF (SWI/SNF) knock-down fibroblasts showed that stimulus-inducible genes are significantly enriched for BAF-dependent activation; BAF-dependent activation of unpatterned inducible promoters is not necessarily accompanied by establishment of a stable NDR.
7. Interestingly, the authors argue that this last observation is due to transient nucleosome displacement/removal at the TSS regions of non-patterned inducible promoters, which enables gene transcription due to continuous BAF activity. In the absence of this activity, nucleosomes reassemble in the transient NDR regions.

As I hope is clear from the above list, this study characterizes nucleosome arrays in many different ways. My lingering impression however is that many of these findings have already been explored in yeast and fly model systems. Specifically, #2 has been extensively discussed in fly (see e.g. Chereji et al., NAR 2016), #3 is also mentioned there. Changes in nucleosome structure following stress have been studied by the Broach, Struhl and Clark groups in yeast, and the idea that some promoters are poised for activation (#4) has not only been discussed at length but also quantitatively modeled. #5 has been widely accepted by many chromatin biologists, based e.g. on various linker lengths observed in human cell types. Studies of how SWI/SNF and other chromatin remodelers affect nucleosome positioning are also pretty standard and go back at least 6-7 years (see e.g. Tolkunov et al. MBC 2011).

In summary, unless a claim can be made that studies in mammalian cells are significantly different from yeast and fly (which does not seem to be the case because most conclusions appear to be the same), the authors need to (i) acknowledge a vast body of previous work and (ii) clearly state what differentiates their work from these earlier studies, and why these advances are significant enough to merit publication in Nat Comm. As it stands, #7 appears to be the most novel contribution but by itself it may not be enough to justify publication in a journal aimed a wide interdisciplinary audience.

Role of cell-type specific nucleosome positioning in inducible activation of mammalian promoters

Agata Oruba¹, Simona Sacconi^{2,1,3} & Dominic van Essen^{2,1,3}

¹ Max Planck Institute for Immunobiology & Epigenetics, Freiburg

² Institute for Research on Cancer & Aging, Nice

³ Equal contributions & corresponding authors: dvanessen@unice.fr; ssacconi@unice.fr

Point-by-point responses to Reviewers' comments:

Reviewer #1 (Remarks to the Author):

This is a clear and well written manuscript from the van Essen group. The van Essen group has a strong background in relating immunobiology (NFκB response) to chromatin structure (primarily HPTMs). In the current manuscript they ask the question: What is the role of nucleosome position at cell type specific inducible promoters? To answer this question they generate high resolution nucleosome position maps by first ChIP-capturing promoter regions (as defined by H3K4me1) followed by MNase digestion to generate nucleosome distribution maps. They use these nucleosome maps to demonstrate that nucleosome organization at inactive inducible promoters may predict cell-type specific response.

This is a good summary of our question, our strategy, and of one of our key conclusions. We also thank reviewer 1 for the kind appraisal of the manuscript.

COMMENTS:

R1.1. Is ChIP_MNase approach comprehensive in its ability to pull down promoter regions? In figure 1.B.ii where the "Proportion of promoters of non-expressed and expressed genes in fibroblasts among all promoters (left) or among those with the 10% highest levels of coverage after enrichment by H3K4me1 ChIP-MNase." It is important here to know what the total numbers of promoters pulled down is. Another way to answer this is what is the total number of rows in 3.A.i.

The numbers of promoters in each group (rows in figure 3ai; 6695 [active], 214 [inducible], 3225 [inactive]) were previously indicated in the legend describing their quantification in fig 3 panel b: we have now moved these numbers to the legend of panel a, so that this is easier to find.

As a direct reply to the initial question, our analysis of the ChIP-MNase using H3K4me1 as a target is not strictly comprehensive, in the sense that not 100% of promoters in each group are pulled-down at the high coverage threshold that we have selected; this is implicit from the quantification of distinct promoter classes bearing each histone modification in figures S1d & e. However, even at the high coverage threshold that we used, we included around 70% of active and inducible promoters, and over 30% of inactive promoters, allowing analysis of a large fraction of promoters belonging to each of these classes. This is now depicted in new figure S1k and we refer to this figure on p3 of the main text.

To make the analysis completely comprehensive, at the expense of reduced coverage at each promoter, an alternative approach would be to set a more lenient coverage threshold for inclusion of promoters. If this threshold is chosen so that the mean promoter coverage level is the same that would be obtained by non-targeted MNase-seq, then all promoters do indeed meet this criterion and would be included (although this approach would obviously not allow the same high-resolution analysis that we have achieved by selecting high coverage promoters, which we consider to be a particular advantage of the H3K4me1

ChIP-MNase strategy). See also the response to point R1.2 below describing more-detailed analysis of enrichment levels.

R1.2. What is the actual enrich at promoter regions? The authors state that "the breadth of analyzed regions is not restricted to the genomic extent of the target mark since it is defined by the initial chromatin fragmentation." It would be reassuring to see that the enrichment works well by calculating the the enrichment at promoters not solely in terms of Fig 1.B.i, but as a calculation of % of Paired end reads that fall within 5KB of a TSS. If the goal of the ChIP enrichment is to reduce the sequence space to generate higher resolution maps then a majority of reads should map to <10% of the genome (or some quantitative global measure of coverage of promoters across the genome).

The cumulative fraction of reads that fall within a specified distance of a TSS are now shown in new fig S1f ii, which indicates that around 40% of reads fall within 10kb (the size of the initial fragmentation for our H3K4me1 ChIP) of an annotated TSS. This set of target intervals corresponds to 16% of the total genome (also shown in fig S1f ii). The remaining reads comprise those that map to H3K4me1-enriched genomic regions other than promoters (which are predominantly enhancers), as well as a low level of background from the H3K4me1 ChIP (see figure S1j for ChIP specificity).

We have also calculated the fraction of reads that map to a defined percentage of the genome, which is now shown in new fig S1f i: this shows that around 60% of reads map to the highest-coverage 10% of the genome (or, similarly, that 6.9% of the genome encompasses half of all reads). As expected, regions surrounding the majority of annotated TSSs are also encompassed within high-coverage regions of the genome.

We now refer to these new analyses & new figure S1f on p3 of the main text.

R1.3. Regarding the "Correction of ChIP-MNase signal for large-scale variation in the magnitude of ChIP recovery." Some acknowledgement should be made that the tradeoff for this type of correction would be to make nucleosome position signal comprable across a diverse set of promoters, but at the loss of occupancy information (see PMID 20683475). This is not a criticism of the approach, rather a clarification of what is being measured. In fact this normalization of occupancy information strengthens the use of the word "position" in the title.

This is a valid point and we have included a sentence in the main text to acknowledge this on p3: *"note that normalization could also mask any kilobase-scale or promoter-to-promoter variations in overall nucleosome occupancy levels."*

R1.4. Regarding "we find that up to 40% of these are marked by elevated DNase sensitivity (figure 2d)" Does this result suggest that 60% of the promoters pulled down in the ChIP-MNase experiment lack a NDR at the TSS? Figure 3.A.i suggests otherwise. or is it that the NDR is not always identified as a DHS?

This is a misunderstanding: we purposely excluded promoters from our analysis of DNase hypersensitive regions (indicated at the start of the paragraph describing these analyses on p6). Promoters were likewise not included in our analysis of DNase hypersensitivity levels at nucleosomal gaps, and we have now amended the main text to indicate this explicitly on p6 "by computationally searching for single-nucleosome-sized gaps *outside promoter regions* within the ChIP-MNase datasets, we find that up to 40% of these are marked by elevated DNase sensitivity".

R1.5. The prose regarding the approach in making Figure 4 is terribly dense and more challenging to understand than it needs to be. Please consider rewriting.

We acknowledge that the section describing figure 4 - "Cell-type-specific nucleosome patterning at inducible promoters can predict responsiveness" - was rather long and unweildy. In part this is due to the concepts that it describes - different ways of clustering or

partitioning the data to derive predictive conclusions - and we feel that it would be difficult to simplify this while describing it accurately.

However, we also realize that we have tried to squeeze two different topics into the same section, and that this may have exacerbated the problem. We have now split these results into two sections by moving the description of 'poised' aspects of patterned promoters (together with some new analyses described below) into a new section: "Patterned inducible promoters are poised for activation" on pp9-10. We hope that this helps to make the overall text more readable.

R1.6. Figure 4.a suggests that a large proportion of active genes are "non-patterned." As nucleosome position patterning is used as a measure of activity and a predictor of responsiveness (because they look like active genes), the unpatterned active genes should be discussed a little bit. Perhaps they are a specific ontologic category? Additionally, are these unpatterned active genes included in the downstream analysis?

This is a very astute point: it is certainly the case that active genes that are inducible in another cell-type, as well as inducible genes after activation, contain a larger fraction of non-patterned promoters than do active genes overall. All of the downstream analyses of active promoters (in figures 2, 3 and 4) include these unpatterned promoters, although they obviously represent only a small fraction of total active promoters. We briefly discussed these promoters in the main text (on p13 and in figure S6f - "the small fraction of non-patterned active promoters ... could represent ... inducible genes that were activated during development, *in vitro* culture, or stimuli that were not purposely controlled in our experiments"), and we hope that it is clear from this that we agree with the notion that these promoters likely represent a distinct class (or classes) of genes.

The suggestion to look for a specific ontologic category is a very good idea, and we have now analysed enrichment of Gene Ontology (GO) terms annotated by the MGI project among non-patterned compared to patterned active promoters.

We first considered the sets of non-patterned active promoters in DCs & fibroblasts that we could define as inducible in the other cell-type based on our experimental stimulation (so, in the case of fibroblasts, corresponding to the non-patterned promoters within the upper [green] group of figure 4a). However, the relatively small number of genes in these sets of promoters (91 genes or less in any set), coupled to the already-high level of enrichment for particular GO terms (dominated by 'immune response', 'response to LPS' and 'response to cytokine', as expected from the experimental stimuli used), did not allow discovery of any further GO terms that were selectively-enriched among the subsets of non-patterned promoters.

We therefore considered non-patterned active promoters from among the full set of active promoters in DCs & fibroblasts (as depicted in figure S6f for fibroblasts). This revealed a number of significantly-enriched annotations for particular biological processes, in line with the notion that non-patterned promoters may encompass genes controlled by distinct regulatory processes.

The GO terms enriched among non-patterned active promoters in DCs include a number of annotations describing cellular responses to extracellular stimuli, as well as components of signalling pathways, suggesting that these may indeed represent inducible genes that respond to stimuli not deliberately tested in our experiments. The GO terms enriched among non-patterned active promoters in fibroblasts prominently include terms related to lipid metabolism, suggesting that these may reflect developmentally-regulated genes linked to the known developmental potential of 3T3 fibroblasts as a pre-adipocytic lineage. However, although the classes of enriched GO terms are suggestive and intriguing - and could be considered to be consistent with our speculation that non-patterned promoters are linked to inducible and/or developmentally-activated genes - we consider that the major

conclusion from this analysis is more simply that non-patterned promoters are indeed enriched for some particular biological pathways.

These results are now presented in new figure S6j, and mentioned in the main text on p13 "*genes characterized by non-patterned active promoters in fibroblasts & DCs are enriched for particular biological pathways consistent with developmental or stimulus-driven activation*".

R1.7. "However, BAF-dependent activation of non-patterned inducible promoters is not accompanied by establishment of a stable NDR or conversion to a patterned nucleosome arrangement resembling other active promoters (figures 5d,e, S5d), and the absence of a stable NDR at non-patterned inducible promoters is unchanged in BAF-knockdown cells (figures 5d,e, S5d)." Given that the remodeling event is temporary, as stated in the next section of the paper, could another likely interpretation of these results be that an NDR is established, but the temporal resolution of the experiment does not allow the observation of the temporary NDR formation (see PMID: 24310001, 26771136).

This is a good point which deserves discussing. At the quoted point in the manuscript, where this result is initially discussed, this suggested interpretation could indeed be valid. We feel that the subsequent experiments presented in figure 6, which investigate the presence of an NDR at selected subsets of promoters at transcribing (pol-II S7P+) or initiating (pol-II S5P+) alleles, argue against this as the main explanation, and that transient NDR formation at initiating alleles is better able to fit the entire data: if promoter activation required formation of a temporary NDR that is stable between cycles of initiation, we would expect to detect this whenever we can recover actively transcribing (pol-II S7P+) alleles; and if our temporal resolution is completely outside the window of promoter activation, we would not expect to recover any pol-II S7P+ alleles of these promoters.

However, we cannot completely rule-out the possibility that temporary formation of stable NDRs may occur at different time-points that we did not assay. In particular, we note that the studies cited by reviewer 1 (Sexton *et al.*, 2014 & 2016) describe temporary formation of NDRs at KSHV-responsive promoters with timescales of 12-24 hours - much longer than the more rapid inducible events we have focused on with timescales of around 1 hour - and it is conceivable that prolonged promoter activation may lead to nucleosome depletion at later time-points.

We have now amended the results and discussion sections of the text to acknowledge this possibility: results p11: "the absence of a stable NDR at non-patterned inducible promoters is unchanged in BAF-knockdown cells *at the time-point analysed*"; discussion p16: "initiation of transcription from these promoters neither requires nor induces formation of a stable NDR *at the timescale analysed*" and "*although we cannot exclude that stable NDRs may temporarily form at time-points not analysed in our experiments, nor that they may later become established after prolonged activation (Sexton 2014; Sexton 2016).*"

R1.8. This reviewer simply does not understand the use of "allele specific" throughout this manuscript, and the language should be tightened up. The expectation set up in the second sentence of the summary is that differential organization at the maternal and paternal alleles of the same gene will be resolved using SNPs from the individual sequencing reads. This is not the case, the use of "allele" throughout this manuscript is confusing.

We refer to ChIP-MNase as 'allele specific' because it can distinguish between different alleles of the same gene, according to features defined by the ChIP target: for instance, it enables specific analysis of transcribing alleles (and not non-transcribing alleles of the same gene) when using pol-II S7P as a ChIP target, or separation of initiating and non-initiating promoter alleles (when using pol-II S5P as a target). We think that this usage matches the most-general meaning of the term 'allele specific'.

However, ChIP-MNase is not '*parental* allele specific', which appears to be what reviewer 1 expects from our use of the term. We are cognizant that 'allele specific' is very frequently

used as a shorthand for '*parental* allele specific', and that discrimination of paternal vs maternal alleles is one of the most commonly-used types of allele-specific analyses. Thus, we agree that it could be worthwhile to 'tighten-up' the language to avoid this kind of confusion, even if we believe that we have used the term in the technically correct way.

Therefore, we have introduced the following changes to the wording of the manuscript:

- Abstract: change 'high-resolution and allele-specific nucleosome profiling at selected genomic features' to '*high-resolution nucleosome profiling at selected genomic features and functionally-defined alleles*'.
- p4: change 'analyse nucleosome positioning at diverse genomic features, and in an allele-specific fashion' to '*analyse nucleosome positioning at diverse genomic features, and at a selectively-marked subset of alleles*'.
- p13: 4th paragraph: 'we estimate that around 9 - 10% of active promoters exhibit allele-specific or initiation-specific nucleosome patterning': leave as-is.
- p13: 5th paragraph: change 'analysis of nucleosome positioning at chosen genomic regions, and in an allele-specific fashion' to '*analysis of nucleosome positioning at chosen genomic regions, and at selectively-marked alleles*'.
- p14: 'nucleosome occupancies are determined in an allele-specific manner': leave as-is.
- p15: change 'techniques that would profit from feature-specific enrichment or allele-specific analysis' to '*techniques that would profit from feature-specific enrichment or selective analysis of functionally-defined alleles*'.

Reviewer #2 (Remarks to the Author):

Oruba et al. present a new method ChIP-MNase. The allure of this method is that it gives an alternative to MNase-seq for large mammalian genomes, where getting good coverage of nucleosomes genome-wide is prohibitively expensive. Their method involves pulling down specific chromatin constituents (H3K4me1 for example) that is part of ~10 kb fragments and then further digestion to mono-nucleosomes of the pull down material. Thus, they are supposed to get a ~10 kb view of nucleosomes around epitope of interest. Using this technique, they seek to map promoter nucleosomes at high-resolution with minimal sequencing, in ground state and stimulated dendritic cells and fibroblasts. First, they observe nucleosome depletion at the promoter and nucleosome patterning akin to active genes at inducible genes even in the absence of stimulation in dendritic cells. Furthermore, they determine the status of inducible genes of dendritic cells in fibroblasts. They observe a set of these genes are active, another set inducible and the rest inactive, as would be expected. Furthermore, the nucleosome depletion status and/or the nucleosome pattern in the gene body is not conserved between the cell types, which strongly suggests that cell type-specific factors are determining nucleosome landscape in the inducible genes (prior to stimulation also). They then tried to deplete BAF using shRNA and observed minimal effect of this knockdown on expressed genes and the inducible genes that had NDRs prior to stimulation. The inducible genes with occluded promoters are affected the most due to BAF knockdown. Finally, they demonstrate ChIP-MNase with Ser5 and Ser7 phosphorylated RNA polymerase II (RNAPII). Here, they find a striking phenomenon where, inducible genes with occluded promoters as observed by H3K4me1 ChIP-MNase, had nucleosome depletion when profiled by initiating RNAPII (Ser5 phosphorylated), indicating that there is transient remodeling of these promoters during initiation.

This is an accurate synopsis of the ChIP-MNase approach, and a good summary of our main results.

Overall, this is a useful method to save sequencing costs, but the presentation needs to be condensed and improved (suggestions below) for a wide audience to appreciate this study. Furthermore, the conclusions regarding determinants of nucleosome depletion at inducible genes can be strengthened based on both prior research and possible analysis with their own datasets.

Specific comments:

R2.1. Page 4: "The requirement for much lower sequencing depths to achieve high coverage across regions-of-interest using ChIP-MNase facilitates the inclusion and analysis of replicate biological samples."

Page 4: "substantially lower levels of sequencing than are required by non-targeted approaches."

The Authors need to present some data regarding the reduction in total sequencing reads possible by ChIP-MNase compared to global MNase. Since this is the main advantage of the method presented here, subjective statements (quoted above) in the text are not enough to prove the utility of the method. Overall, I agree with the Authors that this would be cheaper, but this needs to be presented explicitly and quantitatively.

The reduction in sequencing reads possible by ChIP-MNase (compared to global genomic analysis by non-enriched MNase digestion) is equivalent to the enrichment in coverage at the chosen genomic features-of-interest. This is reported at various points in the manuscript text (summarized here) and presented quantitatively in figures 1b and S1f-i:

- Results p3: "Fragments mapping to regions surrounding annotated promoters were significantly enriched within the H3K4me1 ChIP-MNase datasets, with a mean increase of 3- to 5-fold when compared to the level corresponding to uniform genomic coverage."
- Results p3: "...many promoters exhibited much higher levels of enrichment, with the top 10% (2588 promoters) on average 14- to 16-fold higher than the genome-wide average (figure 1b)".
- Discussion p14: "the increase in coverage was 6- to 35-fold, across 5kb regions, depending on the immunoprecipitating antibody used."
- Discussion p14: "High-levels of coverage are critical in order to define nucleosome positions with accuracy ... (equivalent to ... around 100M reads per genome without enrichment) ... ChIP-MNase allows these levels to be comfortably surpassed using ... typically 20M reads per sample".
- Figure 1b displays "fold-enrichment over the mean level corresponding to homogeneous genomic coverage" of coverage levels attained by ChIP-MNase.
- Figure S1g-i displays "fold-enrichment of sequence coverage by H3K4me1 ChIP-MNase ... compared to the level corresponding to uniform genomic coverage."

In addition, the new figure S1f i now also presents the same data in a different way, by depicting the fraction of the genome that is encompassed by high sequencing read coverage.

We feel that, together, these quantitative measures and figures quite comprehensively document the level of enrichment that can be achieved by ChIP-MNase (using H3K4me1 as a target and initial fragmentation to 5-10kb pieces).

Nevertheless, we realize that the equivalence between enrichment for target regions and the reduction in sequencing reads required was not stated explicitly in the main text, and that this might lead to some ambiguity or confusion in the interpretation of these results. Therefore, we have now amended the main text and figure legends to make this more apparent:

- Results p3: "...a mean increase of 3- to 5-fold when compared to the level corresponding to uniform genomic coverage (*or, put differently, the same level of coverage at these regions was attained from 3- to 5-fold fewer total sequencing reads, compared to non-targeted MNase digestion*)".
- Legend to figure 1b: "the fold-enrichment over the mean level corresponding to homogeneous genomic coverage (*equivalent to the fold-reduction in sequencing depth that would be required to attain a fixed level of coverage*)".
- Legend to figure S1l,m: "Note that the level of coverage mapping to promoters in the whole-genome MNase-seq data analysed here (27 reads kb⁻¹) is lower than the level

attained by ChIP-MNase (≥ 50 reads kb^{-1} at all promoters analysed), *despite a substantially higher total sequencing depth (whole-genome MNase-seq: 192M total sequenced fragments; ChIP-MNase-seq: 37-40M sequenced fragments per replicate)*".

R2.2. For all the following quotes/figures, just the mean values are presented, which preclude readers from evaluating the quality and/or biology revealed by the data. These should be replaced by histograms or violin plots that show the full distributions:

Page 3: "The increased coverage of fragments mapping to ChIP-enriched regions compared to other genomic locations, which serves as an estimate of the specificity of the measured signal for alleles bearing the immunoprecipitated mark, was typically around 30-fold (figure S1i)"

We have now replaced this figure with a new version (new figure S1j) that displays violins indicating the distributions of coverages for each group of regions.

Page 4: "The mean difference in nucleosome positions predicted using ChIP-MNase datasets generated from individual replicate samples is approximately 30bp"

This sentence refers to figure 1f, which displays mean values and error bars indicating the standard errors of the means for differences in predicted nucleosome positions between replicates. We have now added a version of this figure that displays violins indicating the full distributions of absolute differences between replicates as new figure S1q, and we refer to this in the legend to figure 1f.

Page 4: "we measured mean spacings of 192bp in fibroblasts and 183bp in DCs by ChIP-MNase (figure 2a; note that the precision attained by averaging data from millions of mapped fragments can greatly surpass the 30bp resolution of individual nucleosome positions)."

The nucleosomal spacing depicted in figure 2a is calculated for each sample as the period of the full distribution of relative positions of sequence tags, which is directly shown in figures S2a and S2b. These supplemental figures previously depicted only the distribution obtained from DCs; we have now updated them to also show the distribution obtained from fibroblasts.

The nucleosomal period can also be estimated from the distances between predicted nucleosome positions (although this approach inherently discards information about the 'fuzziness' of nucleosomal positioning, and detection of closely-spaced alternative nucleosomal positions is also sensitive to the cut-off selected to define 'presence' and 'absence'). This is visible from the full distribution of inter-nucleosomal spacings based on predicted nucleosome positions in figure S1p (rather than being based on sequence tags as in figures S2a and S2b).

Finally, in response to the question by reviewer 2 below (R2.3), we have also analysed the distribution of internucleosomal spacings between immediately-adjacent predicted nucleosome positions, focusing on the +1 nucleosomes at the TSS. This allows us to plot the distribution of internucleosomal spacings without the contribution from the distances to the next-but-one and next-but-two (and so on) nucleosomes (that were used to calculate the period), and is another intuitive way to depict the distribution of spacings between nucleosomes. This distribution is shown in reviewer-only figure R1c (see also response to point R2.3 below).

R2.3. Page 5: "We find that the 'phased' pattern of mean nucleosome occupancy is largely undiminished both downstream and upstream of the TSS when nucleosomes are individually aligned (figure S2l,m)"

Here, I assume 0 represents the position of the centered nucleosome. The y-axis is cut-off so that we can't even see the centered nucleosome. The adjacent nucleosomes have signal around the mean. This would mean that the probability of observing adjacent

nucleosomes is really low, undermining the claim of the Authors that the MNase-ChIP has alleviated the read depth issue of MNase-seq.

We had not foreseen that figure S2I-n might be interpreted in this way, which - as pointed-out by reviewer 2 - would contradict most of our other data, and we have now carefully looked-into this.

Firstly, reviewer 2 is correct that relative position 0 represents the position of the centred nucleosome. Since all the data are purposely aligned with a local maximum at this position, it exhibits the maximum mean occupancy signal (in the range of 1.7 to 2.4 for all panels); however, this high value is driven primarily by the alignment strategy, rather than by any biological meaning, so we considered that it was sensible to truncate the y-axis signal, in order to allow visualization of the signal corresponding to the adjacent nucleosomes (which is the purpose of the figure). Each plot of mean occupancy is normalized to a mean level of 1 across the entire interval shown, to enable the different plots to be compared to each other: thus the fact that the adjacent nucleosomes show a signal oscillating around the mean is expected, regardless of the probability of observing adjacent nucleosomes. We have amended the legend to figure S2I-n to clarify this.

To directly investigate whether the probability of observing adjacent nucleosomes is indeed low - which would indeed undermine our claim and would seem to contradict most of our other data - we first examined predicted nucleosome midpoints at all promoter classes (as in figure S3b), after sorting promoters in order of the distance separating the predicted +1 and +2 nucleosomes. This clearly reveals that an adjacent +2 nucleosome is detected within a <280bp (1.5x mean nucleosome spacing) distance at the majority of promoters (reviewer-only figure R1a). We also noticed that the small number of active promoters with a predicted +2 nucleosome at a distance of ≥ 280 bp often exhibit reduced nucleosome occupancy levels within this interval in other experimental samples, suggesting that many of these correspond to a true nucleosomal 'gap', rather than a predictive or detection failure.

To address this more quantitatively, we examined the cumulative number of predicted nucleosomes, and, separately, the cumulative probability of detecting at least one nucleosome, within increasing distances from the +1 nucleosome at active promoters (starting at a relative position of +100bp; reviewer-only figure R1b,c). The cumulative number of predicted nucleosomes describes a continuously-increasing, wavy line, with a mean gradient close to that expected from a mean increment of one nucleosome every nucleosomal period, confirming that 'gaps' in the predicted nucleosomal array are rare. The cumulative probability of detecting at least one nucleosome within a given distance directly quantifies the rate of observing adjacent nucleosomes, and indicates that at 90% of promoters, an adjacent nucleosome is detectable within 290bp of the +1 nucleosome, and at 99% of promoters within 450bp. The distribution of distances to the first adjacent nucleosome - equivalent to the gradient of this plot - is overlaid on figure R1c, and represents another way to display the distribution of internucleosomal spacings at this position (see point R2.2 above).

Altogether, these additional analyses confirm that ChIP-MNase is able to consistently detect nucleosomes with high reliability, and figure S2I-n does not disagree with this. We hope that the amended legend describing figure S2I-n will make this sufficiently clearer.

R2.4. Page 6: "Strikingly, we observed that nucleosome patterning at many - but not all - inducible promoters closely resembles that at active promoters, even in non-stimulated cells before induction of gene expression."

Discussion: "Similarly, we cannot unambiguously determine from our data whether basal transcription represents a cause or consequence of nucleosome patterning at some promoters."

The fact that most of these genes have paused Pol2 has to be stated when this statement is made.

Furthermore, presence of paused Pol2 at inducible genes indicates that stimulation might regulate elongation. This is the simplest explanation for presence of NDRs at many inducible genes. The Authors could examine RNAPII-Ser5p data in un-stimulated state to confirm this.

This is an appealing notion that we previously conjectured in the original text (“pre-loading of RNA pol-II before stimulation ... could poise them for inducible activation without requiring stimulus-driven polymerase recruitment” [replaced in new text]).

As suggested by reviewer 2, we have now performed a number of additional analyses to better address this, which are now described in a new section of the text, “Patterned inducible promoters are poised for activation”, on pp9-10, and shown in figure S4l-o. First, we analysed the levels of elongation-competent (S7P-modified) RNA pol-II at patterned and non-patterned promoters separately (figure S4l): this immediately revealed that, while patterned promoters are indeed generally pre-loaded with paused pol-II before stimulation, pre-loaded pol-II can also be found at a many non-patterned promoters. Thus, pol-II binding alone does not appear to be sufficient to explain the division of patterned and non-patterned promoters (although, as suggested, it may nonetheless contribute to NDR formation or stability at some patterned promoters). Secondly, we observed that both classes of promoters exhibit strong, stimulus-induced increases in occupancy by both initiating (S5P) as well as elongating (S7P) RNA pol-II (figure S4n). Hence, stimulation at both patterned and non-patterned promoters regulates both transcriptional initiation as well as elongation. Indeed, even patterned promoters that have high levels of pre-stimulation paused pol-II undergo further increases in promoter pol-II levels upon stimulation (figure S4o), arguing that stimulus-driven initiation occurs at essentially all inducible promoters, and that few - if any - are regulated by elongation *alone*. We note that these results are still consistent with the notion that stimulation may regulate elongation in parallel to new pol-II recruitment (and it would require experiments to selectively block initiation while sparing elongation to definitively address this, which would be out of the scope of our current study), and we have left this possibility open: “inducible gene expression from patterned promoters can reflect both the activation of pre-bound, paused RNA pol-II, as well as stimulus-induced transcriptional initiation” (p10).

The sentence quoted from the discussion (p15), is already part of an entire paragraph that extensively discusses paused RNA pol II, and indeed the preceding 3 sentences all mention the possible link between pol II pausing and promoter patterning: we therefore feel that this topic is adequately covered at this point in the manuscript: “The presence of a pre-stimulation NDR suggests that the TSS at many inducible promoters may be directly accessible for binding by the transcriptional machinery (including the PIC and/or RNA pol-II). Consistent with this, we find that patterned inducible promoters are generally pre-loaded with paused RNA pol-II ... It has been proposed that the presence of paused RNA pol-II at promoters may compete with nucleosome assembly and thereby help to maintain a poised configuration; however, other reports have suggested that aspects of promoter patterning, including the NDR and positioned +1 nucleosome, may instead represent contributory causes of polymerase pausing”.

We hope that with the addition of the new results and section of the text describing the incidence of paused pol-II at patterned and non-patterned promoters, this topic is now adequately well-covered.

R2.5. Page 7: "nor does it arise through preferential recovery of a minority of patterned alleles by the H3K4me1 ChIP-MNase technique (figure S3h-k)."

The Authors should separate patterned induced and non-patterned induced genes to confirm to confirm that they see same feature by MNase also. This would rule out the possibility that they are observing effect of pulling down H3K4me1.

This is indeed an important point, and we had originally attempted to address this in the heatmap of figure S3h by sorting the inducible promoters in order of the measured level of patterning in non-stimulated cells: the clear presence of an NDR and prominent +1 nucleosome at the topmost promoters in this panel illustrates that promoter patterning is also detectable at non-stimulated inducible promoters in whole-genome MNase-seq data, and that this patterning is highly-enriched among the promoters that exhibit the highest patterning levels measured by ChIP-MNase.

We have now also performed the analysis directly suggested by reviewer 2, by separating patterned and non-patterned inducible promoters and analysing the nucleosome occupancy profiles across each set separately, when measured by ChIP-MNase or by whole-genome MNase. This is now shown in new figure S3j, which again confirms that the division of inducible promoters into 'patterned' and 'non-patterned' subsets is not a consequence of the H3K4me1 pull-down used, and cited at the same point in the text on p7.

R2.6. Page 10: "Therefore, establishment and maintenance of a stable NDR at the promoters of most expressed genes does not require the normal remodelling activity of the BAF complex."

The Authors do not show BAF complex protein levels after knockdown. The perdurance of BAF remodelers after knockdown by RNAi could explain the results they observe. One possibility is that BAF complex is knocked down only marginally at the protein level, and the levels are enough for continued function at active genes and inducible genes with NDRs. Whereas, the levels are not enough for BAF function at inducible genes without NDR.

We fully agree with this notion as a possible interpretation of our data, and this was exactly what we had tried to convey when we describe these results on p10 of the original text (*italics added here*): "the activity of the BAF complex is generally non-essential for the majority of steady-state transcription (*or that the low residual levels in knockdown cells are sufficient for this*)".

Our intention was that "the normal remodelling activity of the BAF complex" in the sentence quoted by the reviewer should imply remodelling activity *at its normal level*, in agreement with the interpretation of reviewer 2. We have now adjusted this sentence in the results section on p11 to unambiguously make this meaning clear: "Therefore, establishment and maintenance of a stable NDR at the promoters of most expressed genes does not require *normal levels of remodelling activity by the BAF complex*".

R2.7. A similar approach was described in this paper: <https://www.ncbi.nlm.nih.gov/pubmed/31371436>, which needs to be discussed.

This paper (Comoglio *et al.*, Genes Dev 2019; which was published while our manuscript was under submission), uses ChIP for particular histone modifications to enrich for genomic regions of interest, after MNase digestion. This is an approach which has also been previously used in some other studies which we discuss on p14 of the text. As we originally discussed, a key difference between our ChIP-MNase strategy and these other reports is that we use ChIP to pull-down large chromatin fragments - bearing multiple nucleosomes - and subsequently analyse nucleosome positioning on them independently from their pattern of modifications. This allows us to separate changes in nucleosome positioning from changes in the levels of histone modifications on individual nucleosomes, which is not possible when the steps are performed the other way around. Direct analysis by ChIP of previously MNase-digested chromatin also renders the use of non-histone ChIP targets difficult, since these are frequently lost during the MNase digestion. See also our reply to reviewer 3 below (R3.1)

We have now added citations to Comoglio *et al.*, at this point in the discussion on p14: "ChIP of MNase-digested chromatin has also been used to selectively analyse the distribution of nucleosomes bearing defined histone modifications studies (Comoglio *et al.*, 2019; Scruggs *et al.*, 2015; Wal *et al.*, 2012; Weiner *et al.*, 2010; Barski *et al.*, 2007; Li *et*

al., 2011; Mavrich *et al.*, 2008; Schmid *et al.*, 2007; Zhang *et al.*, 2008); however, this approach is unable to directly distinguish between changes in nucleosome occupancies and changes in histone modification levels (Barski *et al.*, 2007; Li *et al.*, 2011, Comoglio *et al.*, 2019), and it cannot be readily used to map nucleosome positions surrounding non-histone targets”.

Reviewer #3 (Remarks to the Author):

First of all, I would like to emphasize that this is a very thorough and well-described study of nucleosome positioning in mammalian cells. The authors correctly point out that previous studies of this type have been somewhat hindered by low sequencing depths and the absence of replicates. The authors circumvent this difficulty by employing what they call ChIP-MNase, with the H3K4me1 histone modification as the ChIP target. Thus they trade genome-wide coverage for a higher sequencing depth, at the risk of creating a biased sample (the authors argue that H3K4me1 is safe in this regard because it is found at both active and inactive promoters and I would tend to agree with this assertion).

This is a good summary of the benefits and the associated risk of the ChIP-MNase approach. We indeed argue that the use of H3K4me1 as a ChIP target is valid for the analyses that we have performed. To say that it is ‘safe’ would probably be an over-statement, and we have tried to avoid making this claim: instead, we believe that through the various controls in figures 1b ii, S1c-m (now including new figure S1f) and S3h-j (now including new figure S3j), we have demonstrated that it is an appropriate ChIP target to address the questions that we have tackled. However, we hope that it is clear from our manuscript that we did not take this ‘for granted’, and that these were important controls.

That said, we appreciate that reviewer 3 agrees with the validity of our use of H3K4me1.

R3.1. Although the ChIP-MNase technique is described as novel in the manuscript, it is clearly a combination of well-known methods for exploring chromatin structure. Similar protocols have been developed in Struhl, Rando, Broach labs and probably many others.

We certainly agree that ChIP-MNase is based on a combination of two well-known methods - ChIP and MNase-digestion - and we hope that our use of a straightforward name to describe the technique makes this immediately clear.

We also acknowledge that many other labs have developed variants of the MNase-based nucleosome-mapping strategy, and we do not intend to diminish their importance: we previously included a brief overview of some of these in the discussion section of the manuscript on p14, and we have now updated this with a more extensive set of references.

In particular, and as also discussed above (reply to reviewer 2 point R2.7), previous published studies have performed ChIP for particular histone modifications using MNase-generated mononucleosomal fragments, in order to map nucleosomes at high coverage across specific genomic regions. However, a fundamental limitation of this approach is that only nucleosomes bearing the target histone modification are analysed: thus, nucleosome positioning can only be inferred within the genomic extent of the target modification, and it is difficult or impossible to discriminate between the level (or absence) of nucleosome occupancy at a particular genomic position and the level (or absence) of the targeted histone modification: this limitation has been noted previously by other groups, and we have also added citations to these to the relevant point in the text on p14 (Barski *et al.*, 2007; Li *et al.*, 2011; Comoglio *et al.*, 2019). In contrast, our approach of first crosslinking nucleosomes in place on large chromatin fragments, and then performing ChIP to enrich for genomic features-of-interest before determining nucleosome positions by subsequent MNase digestion, avoids both of these problems, and also enables the possibility of selectively mapping nucleosomes around non-histone chromatin features.

Thus, although it is absolutely correct to describe ChIP-MNase as simply 'a combination of well-known methods', we feel that this particular setup represents a key conceptual difference to previous approaches.

R3.2. The authors carry out a thorough analysis of nucleosome maps under various conditions. Their main findings include:

1. Observation of different internucleosome linker lengths in 3T3 vs. DC cells.
2. Nucleosome "patterning" or periodicity is much more pronounced in active vs. inactive promoters, with the lack of patterning in the latter attributed to the lack of phasing between periodic nucleosome arrays in different inactive genes.
3. Observation of nucleosome depletion at DNase hypersensitive sites.
4. Stimulus-inducible promoters sometimes display patterning of active promoters even before stimulus-driven activation (ie, they are "primed" to be induced, potentially through pre-loading of RNA PolII before stimulation).
5. Nucleosome positioning at many inducible promoters is not dictated by DNA sequence alone (this is already evident from the significantly different nucleosome spacing in 3T3 vs. DC cells).

We would like to interject here to note that the different spacing that we describe between DCs and fibroblasts, and which other studies have reported in different cell types, applies to overall nucleosome positioning throughout the datasets, rather than being restricted to promoter regions. Moreover, inter-nucleosome spacing contributes only a limited degree to the overall promoter patterning that we quantify: the presence/absence and measured magnitude of an NDR, as well as the positioning and occupancy of the +1 nucleosome, each have a more pronounced impact (figure S2g-i) and they are unrelated to inter-nucleosomal spacing. Thus, although the observed differences in overall nucleosome spacing indeed confirm that nucleosome positions cannot be *globally* dictated by DNA sequence alone (and this has been previously shown by other groups), it does not automatically follow from this that the characteristic patterning of nucleosome positions *at promoters* could not be driven by their DNA sequence.

See also our reply below (R3.5).

6. Using BAF (SWI/SNF) knock-down fibroblasts showed that stimulus-inducible genes are significantly enriched for BAF-dependent activation; BAF-dependent activation of unpatterned inducible promoters is not necessarily accompanied by establishment of a stable NDR.

7. Interestingly, the authors argue that this last observation is due to transient nucleosome displacement/removal at the TSS regions of non-patterned inducible promoters, which enables gene transcription due to continuous BAF activity. In the absence of this activity, nucleosomes reassemble in the transient NDR regions.

Points 1-7 are a fairly accurate list of the main findings in our study (although not all are claimed by us to be novel: see below).

We also believe that the discovery and characterization of a subset of inducible promoters which lack active 'patterning' even when activated - and so differ from the vast majority of non-inducible active promoters - represents an important additional finding not listed here (although implicit for points 6 & 7).

As I hope is clear from the above list, this study characterizes nucleosome arrays in many different ways. My lingering impression however is that many of these findings have already been explored in yeast and fly model systems.

Although we feel that many aspects of our results go well beyond previous studies (see below), we have nevertheless updated the title of our study to clarify that we are addressing

nucleosome positioning in the mammalian system: “Role of cell-type specific nucleosome positioning in inducible activation of *mammalian* promoters”.

R3.3. Specifically, #2 has been extensively discussed in fly (see e.g. Chereji *et al.*, NAR 2016), #3 is also mentioned there.

This is correct, but we should point-out that the major findings of points 1,2 & 3 are acknowledged in the manuscript to be already-described, and were primarily used to confirm that the ChIP-MNase approach is able to recapitulate known aspects of nucleosome positioning.

We cited previous studies for each of points 1, 2 & 3 in the results section of the text. We thank reviewer 3 for pointing-out the paper by Chereji *et al.*, and we have now also included this citation at the relevant places.

Point 1 (p4): “Previous studies using single samples of MNase-digested chromatin have reported a similar observation (Valouev *et al.*, 2011; Teif *et al.*, 2012)”.

Point 2 (p5) “these observations agree well with many previous data (Chereji *et al.*, 2016; Lai *et al.*, 2017), and demonstrate that ChIP-MNase is able to detect known features of promoter architecture”.

Point 3 (p6): “Mean nucleosome occupancy is depleted at DNase-hypersensitive sites, and the average degree of depletion correlates with the magnitude of DNase sensitivity (Chereji *et al.*, 2016; Grossman *et al.*, 2018).

The section describing these results concludes on p6 with the summary “Collectively, these global analyses confirm that ChIP-MNase is readily able to *recapitulate known aspects of nucleosome positioning at individual genomic sites*, and that the high resolution afforded at specific targeted regions can reveal additional insights”, which we hope makes clear that we are not attempting to claim priority for most of these phenomena.

R3.4. Changes in nucleosome structure following stress have been studied by the Broach, Struhl and Clark groups in yeast, and the idea that some promoters are poised for activation (#4) has not only been discussed at length but also quantitatively modeled.

Firstly, we would like to point out here that ‘changes in nucleosome structure following stress’ - as has been well-described for stress-responsive yeast promoters - is not the general pattern of behaviour that we find at mammalian inducible promoters. In fact, we find that in almost all cases “nucleosome patterning at inactive inducible promoters was largely unchanged upon stimulus-driven activation” (p7). This was not an expected result, and may point to differences in inducible promoter regulation between yeast and mammals. We have tried to highlight this on p7 - “non-patterned inducible promoters lack a clearly-measurable NDR ... and - unexpectedly - this overall configuration persists even after stimulation and activation of transcription” and again in the discussion on p16 “This behaviour ... is also markedly different from the reported behaviour of inducible stress-response genes in yeast”.

We feel that this result illustrates why we believe that the effort of analysing nucleosome positioning at high resolution in mammalian cells is worthwhile, even if similar approaches have already been performed in other organisms with smaller genomes; and - as this example shows - conclusions obtained in other species cannot always be assumed.

Secondly, we acknowledge that the notion of promoter ‘poising’ for activation is well established, and that many aspects of this have been described in previous publications. We cited several studies using multicellular models (*Drosophila* or mammals) in the discussion (on p15: “patterning appears to be associated with ‘priming’ of some promoters ... (Schones *et al.*, 2008; Ramirez-Carrozzi *et al.*, 2009)” and “paused RNA pol-II at promoters may ... help to maintain a poised configuration (Schones *et al.*, 2008; Gilchrist *et al.*, 2010; Gilchrist *et al.*, 2008)” and p16: “NDR size ... was linked to the magnitude of

inducible gene transcription (Scruggs *et al.*, 2015)"). We have now also included several citations in the new section 'Patterned inducible promoters are poised for activation' (pp9-10): on p9: "the presence of a pre-stimulation NDR could facilitate or enable stimulus-inducible activity in a particular cell-type ... This has previously been proposed as a likely prerequisite for inducible gene activation at CGI-promoters (Ramirez-Carrozzi *et al.*, 2009)"; on p10: "nucleosome patterning at this subset of promoters ... imply that they are 'poised' for inducible activation (Adelman *et al.*, 2009; Gilchrist *et al.*, 2010; Gilchrist *et al.*, 2008; Hargreaves *et al.*, 2009; Ramirez-Carrozzi *et al.*, 2009; Saccani & Trabucchi, 2015) and/or for high-magnitude transcriptional output (Scruggs *et al.*, 2015)" and "The presence of paused RNA pol-II ... might enable rapid transcriptional activation ... as has been previously described (Adelman *et al.*, 2009)".

However, we believe that the pre-existence of a patterned nucleosome configuration at a large fraction of inducible promoters before activation represents a separate feature of promoter 'poising' which has not (to our knowledge) been systematically characterized before at individual mammalian promoters. Our data shows that patterned inducible promoters are generally also 'poised' by other criteria - including the presence of H3K4 di- and tri-methylation, and pre-loading of paused RNA pol-II - but also that these features can also be found at a fraction of non-patterned promoters. Thus, promoter nucleosome organization represents a distinct aspect of promoter 'poising' that can occur alongside, rather than as a consequence of, these other previously-described phenomena.

R3.5. #5 has been widely accepted by many chromatin biologists, based e.g. on various linker lengths observed in human cell types.

Note our comment above (R3.2) that global differences in mean linker lengths does not automatically imply differences in nucleosome patterning at promoters, and also that even within a single cell-type, the nucleosome repeat lengths at different genomic features (including promoters) can vary substantially from the mean repeat length (Valouev *et al.*, 2009; Teif *et al.*, 2012 & 2018; also see our figure S2c).

But, to more directly reply to reviewer 3's point, we concur that there is general acceptance that *global* nucleosome positioning is not dictated by DNA sequence alone. However, previous publications in both yeast (Sekinger *et al.*, 2005; Zhang *et al.*, 2009, 2010; Kaplan *et al.*, 2010; Radman-Livaja & Rando 2010; Hughes *et al.*, 2012; Struhl & Segal, 2013) and mammals (Valouev *et al.*, 2011; Fenouil *et al.*, 2012; Barozzi *et al.*, 2014) have argued that promoter DNA sequences can strongly favour NDR formation, and in some cases also contribute to +1 and -1 nucleosome positioning (Valouev *et al.*, 2011). Thus, we feel that the role played by DNA sequence in determining promoter nucleosome positions is still a rather open issue. To set our findings into this context better, we have now added this set of citations at the relevant point in the main text of the manuscript on p7.

Moreover, while differences in bulk nucleosome occupancies between active and inactive genes have been well-studied, cell-type specific differences in nucleosome positioning are particularly pertinent at promoters of stimulus-inducible genes, since these genes are inactive in non-stimulated cells, and so their transcriptional status is often not distinguishable between a cell-type in which they are inducible, and another cell-type in which they are non-expressed and non-inducible. To our knowledge, nucleosome positioning at stimulus-inducible promoters has not previously been studied in distinct cell-types at high resolution.

R3.6. Studies of how SWI/SNF and other chromatin remodelers affect nucleosome positioning are also pretty standard and go back at least 6-7 years (see e.g. Tolkunov *et al.* MBC 2011).

We think that this is not a fair assessment of the published literature describing how BAF (SWI/SNF) affects nucleosome positioning during promoter activation, and particularly not in non-yeast organisms (the paper cited by Tolkunov *et al.* describes yeast SWI/SNF).

It is certainly true that there are many *in vitro* studies that describe the enzymatic function of BAF complexes and other related nucleosome remodellers. However, only a handful of reports using multicellular organisms have addressed the role of BAF remodelling on *in vivo* nucleosome positioning, and (to our knowledge) there are no published studies that describe BAF-dependent nucleosome positioning during inducible promoter activation.

Notably, previous studies in yeast have shown that SWI/SNF and the related RSC nucleosome remodeller play an essential role in the establishment of NDRs at active gene promoters - which is very different from the results that we describe in mammalian cells, where promoter patterning is generally unchanged after BAF-knockdown. We have added a more-complete set of citations to the text of the manuscript (on page 10: Gutierrez *et al.* 2007; Hartley *et al.*, 2009; Tolkunov *et al.*, 2011; Rawal *et al.*, 2018) to set our experiments better into the context of these previous studies.

The few non-yeast studies (that we are aware of) that directly address the role of BAF remodelling on *in vivo* nucleosome positioning are Shi *et al.* (2014), who examined the overall nucleosome landscape at low resolution in Brahma-knockdown *Drosophila* cells; Hu *et al.* (2011), who analysed the nucleosome landscape surrounding GATA1-bound enhancers in Brg1-knockdown haematopoietic cells; and Tolstorukov *et al.* (2013), who analysed mean nucleosome occupancies across promoters in Brg1- and Snf5-knockout fibroblasts. None of these studies - nor, to our knowledge, any others - address BAF-dependent nucleosome positioning during inducible gene activation. We suspect that this may be at least partially due to the prohibitive levels of sequencing that would be required (without enrichment) to map nucleosome positions at high resolution in multiple experimental conditions, and this is one reason why we believe that the ChIP-MNase approach is well-suited to address this.

It is also worth reiterating here that our finding that BAF-dependent activation of non-patterned inducible promoters does *not* generate a stable NDR, and instead that nucleosome depletion occurs only transiently at initiating alleles, was unforeseen, and could not be extrapolated from earlier studies in yeast or other organisms.

R3.7. In summary, unless a claim can be made that studies in mammalian cells are significantly different from yeast and fly (which does not seem to be the case because most conclusions appear to be the same), the authors need to (i) acknowledge a vast body of previous work and (ii) clearly state what differentiates their work from these earlier studies, and why these advances are significant enough to merit publication in *Nat Comm*.

We hope that we can do both of these things.

In general, we believe that describing the behaviour of nucleosomes at mammalian promoters is important, irrespective of whether some aspects turn-out to be the same as those found in organisms with smaller genomes. Indeed, the fact that not all aspects are identical indicates that it is essential to experimentally determine which processes are shared, and which are specific to mammalian promoters. Even if no differences were uncovered (which is not the case) we think that this would represent a significant insight.

We agree with the need to acknowledge earlier work - and we have now incorporated more extensive citations into the manuscript, as detailed in our replies above.

We have also attempted throughout the manuscript to clearly state where our results are in line with previous findings in other systems (for instance, much of points 1-3 above) and where they diverge (for instance, our finding that promoter nucleosome positions are largely unchanged upon inducible gene activation, in contrast to the changes that have been described at stress-responsive genes in yeast; and our finding that the normal activity of the BAF complex is not required to establish stable NDRs or other aspects of patterning at promoters, different to the reported roles of the related SWI/SNF and RSC remodelling complexes in yeast).

In addition to this, our studies have uncovered a number of phenomena that have not previously been characterized. In particular, we feel that the existence of two distinct classes of inducible promoters, separable by their nucleosome patterning; the finding that pre-stimulation nucleosome patterning at inactive inducible promoters can be predictive of cell-type specific responsiveness to stimulation; the finding that BAF-dependent gene expression is strongly biased to regulate stimulus-inducible genes; and the discovery of transient remodelling at initiating alleles of non-patterned promoters (below), are all significant advances.

R3.8. As it stands, #7 appears to be the most novel contribution but by itself it may not be enough to justify publication in a journal aimed a wide interdisciplinary audience.

We agree that point 7 - the transient nature of NDRs at non-patterned inducible promoters - is a key novel aspect of our work, and it is also a finding which could only be made through the allele-specific ability of ChIP-MNase to selectively analyse the subset of initiating (pol-II S5P-bearing) promoter alleles.

As argued above, we also feel that the discoveries outlined in the response to R3.7 above represent novel findings from our study, and provide important insights into the role of nucleosome positioning at inducible promoters.

Altogether, we believe that these new results, together with firmly-establishing principles of nucleosome organization at mammalian promoters that have only been previously shown in very different organisms, will be of interest to a wide audience.

REVIEWERS' COMMENTS:

Reviewer #1 (Remarks to the Author):

The responses to my initial comments are thoughtful and thorough. The authors have provided compelling answers and necessary revisions. I am satisfied with the manuscript in its current state.

-Jonathan Dennis

Reviewer #2 (Remarks to the Author):

My comments were addressed.

Reviewer #3 (Remarks to the Author):

I agree with the authors that their manuscript is a thorough study of nucleosome positioning in mammals, and that such studies are as a rule far less numerous than those in yeast and fly. I also agree that studies of nucleosome positioning in mammals are valuable in their own right, even if the conclusions turn out to be largely the same (which is not quite the case here). Indeed, it would not be justified to simply extrapolate the rules of in vivo nucleosome positioning from yeast to human! (well, if nucleosome positioning were dictated **solely** by sequence then the nucleosome positioning rules would be universal and fully transferable between organisms, but I think most researchers agree at this point that this is not the case).

Even after reading the authors' reply, I am not prepared to concede that the ChIP-MNase technique is truly novel and represents a "key conceptual advance". However, this method is well-suited to the questions of nucleosome positioning in mammalian cells that the authors set out to explore in the first place, and this is what counts in the end.

Finally, in R3.7 of the rebuttal letter the authors provide a nice summary of their major novel findings: "our studies have uncovered ... are all significant advances". However, I was surprised to see that these significance and novelty statements are only partially reflected in the abstract. I would advise the authors to update the abstract by including a stronger statement about mammalian cells (beyond just changing one word in the title) and by describing BAF-related findings more explicitly (which would mention BAF by name). In fact, the statement in R3.7 mentioned above would provide an ideal starting point.

In summary, I am happy overall with the revisions and the explanations provided by the authors. They have done a much better job in citing previous work than before, and at least in the rebuttal letter they have clearly explained what the differences are between their findings and previous work. What still remains to be done is to clearly explain this in the manuscript itself, definitely in the abstract but potentially also in the Introduction, to set up the state-of-the-art for the main questions addressed in this paper. This would also be a good place to say more about the novelty of the ChIP-MNase technique and its suitability to the problems at hand.

Role of cell-type specific nucleosome positioning in inducible activation of mammalian promoters

Agata Oruba¹, Simona Sacconi^{2,1,3} & Dominic van Essen^{2,1,3}

¹ Max Planck Institute for Immunobiology & Epigenetics, Stübeweg 51, Freiburg D79108, Germany

² Institute for Research on Cancer & Aging, Nice (IRCAN), 28 Avenue Valombrose, Nice 06107, France

³ Equal contributions & corresponding authors: dvanessen@unice.fr; ssacconi@unice.fr

Response to reviewers

Reviewer #3 (Remarks to the Author):

I agree with the authors that their manuscript is a thorough study of nucleosome positioning in mammals, and that such studies are as a rule far less numerous than those in yeast and fly. I also agree that studies of nucleosome positioning in mammals are valuable in their own right, even if the conclusions turn out to be largely the same (which is not quite the case here). Indeed, it would not be justified to simply extrapolate the rules of in vivo nucleosome positioning from yeast to human! (well, if nucleosome positioning were dictated *solely* by sequence then the nucleosome positioning rules would be universal and fully transferable between organisms, but I think most researchers agree at this point that this is not the case).

Even after reading the authors' reply, I am not prepared to concede that the ChIP-MNase technique is truly novel and represents a "key conceptual advance". However, this method is well-suited to the questions of nucleosome positioning in mammalian cells that the authors set out to explore in the first place, and this is what counts in the end.

Note that we did not use the phrase 'key conceptual advance' anywhere in the manuscript, nor in the initial response to reviewers.

In this final revision of the manuscript, we describe the ChIP-MNase technique as follows: "we feel that ChIP-MNase represents a flexible alternative approach for selective analysis of nucleosomes at defined genomic features and alleles." which we hope is sufficiently modest.

In any case, we are pleased that reviewer 3 agrees that the method is well-suited to address the questions that we have asked.

Finally, in R3.7 of the rebuttal letter the authors provide a nice summary of their major novel findings: "our studies have uncovered ... are all significant advances". However, I was surprised to see that these significance and novelty statements are only partially reflected in the abstract. I would advise the authors to update the abstract by including a stronger statement about mammalian cells (beyond just changing one word in the title) and by describing BAF-related findings more explicitly (which would mention BAF by name). In fact, the statement in R3.7 mentioned above would provide an ideal starting point.

In summary, I am happy overall with the revisions and the explanations provided by the authors. They have done a much better job in citing previous work than before, and at least in the rebuttal letter they have clearly explained what the differences are between their

findings and previous work. What still remains to be done is to clearly explain this in the manuscript itself, definitely in the abstract but potentially also in the Introduction, to set up the state-of-the-art for the main questions addressed in this paper. This would also be a good place to say more about the novelty of the ChIP-MNase technique and its suitability to the problems at hand.

We have endeavoured to better highlight our key findings within the manuscript, while shortening the text to comply with the overall length limitation. To summarize:

- We have ensured that our major findings - as outlined in R3.7 of the original response to reviewer 3 - are specifically covered in the text, as follows:

1. “the existence of two distinct classes of inducible promoters, separable by their nucleosome patterning” -

This is referred to (unchanged) in the abstract (“a subset of inducible promoters can be activated without stable nucleosome depletion from their transcription start sites”), in the new section title in the ‘Results’ (“Nucleosome patterning defines two inducible promoter classes”), and within the ‘Results’ text (“inducible promoters are heterogeneous and can be divided into two groups”),

2. “the finding that pre-stimulation nucleosome patterning at inactive inducible promoters can be predictive of cell-type specific responsiveness to stimulation” -

This is referred to (unchanged) in the abstract (“the nucleosome profile at many inactive inducible promoters is sufficient to predict cell-type specific responsiveness”), in the new summary paragraph at the end of the ‘Introduction’ (“We find that nucleosome patterning at inducible promoters can define their cell-type-specific responsiveness”), in the new section title in the ‘Results’ (“Nucleosome patterning can predict promoter responsiveness”), and within the ‘Results’ text (“the level of patterning in fibroblasts before stimulation was strongly indicative of their subsequent responsiveness”),

3. “the finding that BAF-dependent gene expression is strongly biased to regulate stimulus-inducible genes” -

This is referred to (unchanged) in the ‘Results’ section (“BAF-driven nucleosome remodelling is particularly required for regulated - rather than ubiquitous - gene activation”) and in the ‘Discussion’ (“the remodelling activity of the BAF complex ... [is] strongly enriched among stimulus-inducible genes”),

4. “the discovery of transient remodelling at initiating alleles of non-patterned promoters” -

This is referred to (unchanged) in the abstract (“These promoters ... exhibit transient nucleosome depletion only at alleles undergoing transcription initiation”), in the new summary paragraph at the end of the ‘Introduction’ (“activation of non-patterned inducible promoters is associated with short-lived nucleosome remodelling events that accompany transcriptional initiation”), and (unchanged) in the ‘Results’ (“activation of these promoters is associated with transient NDR formation at initiating alleles”) and ‘Discussion’ (“transcription initiation is linked to mainly transient remodelling events”).

- We have retained almost all of the citations to previous work that were added during the review process;

- We have unfortunately not been able to further ‘set up the state-of-the-art’ or to ‘say more about the novelty of the ChIP-MNase technique’ due to length restrictions, but we hope that the reviewer and editors will agree that the novelty of the technique should be apparent from the text and from the results.